# Collaborative Learning of Discrete Distributions under Heterogeneity and Communication Constraints

**Xinmeng Huang** [*],    **Donghwan Lee** [*]
Graduate Group in Applied Mathematics and Computational Science
University of Pennsylvania
Philadelphia, PA 19104
{xinmengh, dh7401}@sas.upenn.edu

**Edgar Dobriban**
Department of Statistics and Data Science
University of Pennsylvania
Philadelphia, PA 19104
dobriban@wharton.upenn.edu

**Hamed Hassani**
Department of Electrical and Systems Engineering
University of Pennsylvania
Philadelphia, PA 19104
hassani@seas.upenn.edu

## Abstract

In modern machine learning, users often have to collaborate to learn distributions that generate the data. Communication can be a significant bottleneck. Prior work has studied homogeneous users—*i.e.*, whose data follow the same discrete distribution—and has provided optimal communication-efficient methods. However, these methods rely heavily on homogeneity, and are less applicable in the common case when users' discrete distributions are heterogeneous. Here we consider a natural and tractable model of heterogeneity, where users' discrete distributions only vary sparsely, on a small number of entries. We propose a novel two-stage method named SHIFT: First, the users collaborate by communicating with the server to learn a central distribution; relying on methods from robust statistics. Then, the learned central distribution is fine-tuned to estimate the individual distributions of users. We show that our method is minimax optimal in our model of heterogeneity and under communication constraints. Further, we provide experimental results using both synthetic data and $n$-gram frequency estimation in the text domain, which corroborate its efficiency.

## 1 Introduction

Research on learning from data distributed over multiple computational units (machines, users, devices) has grown in recent years, as data is commonly generated by multiple users, such as smart devices and wireless sensors. While many works focus on learning predictive models with distributed data, learning the data distribution itself is also increasingly popular [56, 3, 14, 15, 60]. In various applications, it is often required to reconstruct the data distribution from scattered measurements. Examples include sensor networks and P2P (Peer2Peer) systems, load balancing and query processing (see, *e.g.*, [41, 61, 46] and references therein). In these scenarios, communication costs and bandwidth are often bottlenecks on the performance of learning algorithms [22, 5, 29, 17]. The bottlenecks become even more severe in federated analytics [31], where many users coordinate with a server to learn central models, while communication via the wireless links is typically expensive and operates at low rates.

---

[*]Equal Contribution

36th Conference on Neural Information Processing Systems (NeurIPS 2022).

Table 1: Estimation error $\mathbb{E}[\|\widehat{\mathbf{p}}^t - \mathbf{p}^t\|_2^2]$ of various methods when $n$ is sufficiently large: $\mathbf{p}^t$ is the test distribution, $\widehat{\mathbf{p}}^t$ is the estimator, $\boldsymbol{\delta}^t = T^{-1} \sum_{t' \in [T]} \mathbf{p}^{t'} - \mathbf{p}^t$ is a non-vanishing measure of heterogeneity. See Section 1.1 for other notations. Constants and logarithmic factors are omitted for clarity. The "data usage" column indicates whether the estimate is obtained for each cluster separately or by pooling data.

| Method | Estimation Error | Data Usage | Bound Type |
|---|---|---|---|
| Unif. Group./Hash. [26] | $O\left(\frac{d}{2^b n}\right)$ | Separate | Upper |
| Unif. Group./Hash. [26] | $O\left(\|\boldsymbol{\delta}^t\|_2^2 + \frac{d}{2^b T n}\right)$ | Pool | Upper |
| Localize-then-Refine [15] | $O\left(\frac{\|\mathbf{p}^t\|_{1/2}}{2^b n}\right)^*$ | Separate | Upper |
| Localize-then-Refine [15] | $O\left(\|\boldsymbol{\delta}^t\|_2^2 + \frac{\|\frac{1}{T}\sum_{t' \in [T]} \mathbf{p}^{t'}\|_{1/2}}{2^b T n}\right)^*$ | Pool | Upper |
| SHIFT (Theorem 1) | $\tilde{O}\left(\frac{\max\{2^b, s\}}{2^b n} + \frac{d}{2^b T n}\right)$ | – | Upper |
| SHIFT (Theorem 3) | $\Omega\left(\frac{\max\{2^b, s\}}{2^b n} + \frac{d}{2^b T n}\right)$ | – | Lower |

$^*$ This method [15] requires interactive communication protocols, while other methods are non-interactive.

This paper considers learning high-dimensional discrete distributions from user data in the distributed setting. Many communication-efficient methods have been proposed, and their optimality under communication constraints has been established under various models [26, 6, 27, 2, 3, 14, 15]. However, the key challenge of *heterogeneity*, *i.e.*, that users' distributions can differ, is rarely considered. Heterogeneity is common, as users inevitably have unique characteristics [13]. Meanwhile, heterogeneity can cause a significant performance drop for learning algorithms designed only for i.i.d data [37, 34, 20, 59]. To use all the data, one needs to learn some central structure, transferable to all individual users. Then one may locally learn the unique components for each user [21, 53, 16, 57].

To study this paradigm, we first need to introduce a suitable model of heterogeneity. We consider, as an example, the heterogeneous frequencies of words across different texts, *e.g.*, news articles, books, plays (tragedies and comedies), viewed as users. Most words appear with almost the same probabilities in different texts, however, a few can have very different probabilities, such as "sorrowful" being common in tragedies and "convivial" being common in comedies. Motivated by this, we formulate a model of sparse heterogeneity. Specifically, suppose that the discrete distributions of all users differ from an underlying central distribution in at most $s$ entries, where $s$ is much smaller than the dimension $d$. Sparse heterogeneity is relevant to applications such as recommendation systems [28, 42, 35, 7] and medical risk scoring [50, 43, 40].

However, given data generated by multiple distributions with sparse heterogeneity, previous works [26, 2, 3, 15] either do not use all the data, or suffer from bias due to heterogeneity that does not vanish as the sample size increases. Here we propose a novel **s**parse **h**eterogeneity **i**nspired collaboration and **f**ine-**t**uning method (SHIFT) where we first collaboratively learn the central distribution, and then fine-tune the central estimate to individual distributions. Our method makes full use of heterogeneous data, leading to a significant improvement in error rates compared to prior methods. See Table 1 for an overview, explained in detail later.

## 1.1 Contributions

We consider the problem of learning $d$-dimensional distributions with $s$-sparse heterogeneity. We assume there are $T$ clusters of user datapoints, and allow each datapoint to be transmitted with $b$ bits of information to the server. Our setting embraces heterogeneous data and thus is a significant generalization of the models from [26, 6, 27, 2]. Our technical contributions are as follows:

- We propose the SHIFT method to learn heterogeneous distributions with collaboration and tuning, in a sample-efficient manner. Our method can in principle be used with an arbitrary robust estimate for the probability of each entry/coordinate. When entry-wise median and trimmed mean are used, we provide upper bounds on the estimation error of individual distributions in the $\ell_2$ and $\ell_1$ norms. We show a factor of $\min\{T, d/\max\{s, 2^b\}\}$ improvement in sample complexity compared to previous works; showing the benefit

of collaboration (large $T$) and sparsity (small $s$), despite communication constraints and heterogeneity.

- To justify the optimality of our method, we prove minimax lower bounds on the estimation errors for individual distributions in the $\ell_2$ and $\ell_1$ norms, holding for all, possibly interactive, methods. These lower bounds, combined with our upper bounds, imply that our median-based method is minimax optimal.
- We support our method with experiments on both synthetic and empirical datasets, showing a significant improvement over previous methods.

## 1.2 Related Works

**Learning with heterogeneity.** Learning with heterogeneity is commonly found in the broader context of multi-task learning [12, 52, 8] and federated learning [4, 44, 24, 47], where a central model or representation is learned from multiple heterogeneous datasets. These central representations can be useful for few-shot learning tasks [51, 23] due to their ability to adapt to new tasks efficiently. In heterogeneous linear regression, [53, 21] show improved sample complexities by assuming a low dimensional central representation, compared to the i.i.d. setting [24, 36]. Related results are proved by [16] for personalized federated learning. [57] study a bandit problem where the unknown parameter in each dataset equals a global parameter plus a sparse instance-specific term. We study a different setting: learning distributions with sparse heterogeneity under communication constraints.

**Estimating distributions under communication constraints.** Estimating discrete distributions has a rich literature [10, 18, 45, 19]. Under communication constraints, [26, 6, 27, 2] consider the non-interactive scenario and establish the minimax optimal rates, in terms of data dimension and communication budget, via potentially shared randomness, when all users' data is homogeneous. The optimality for the general interactive (blackboard) methods is developed by [1]. A few works study the estimation of sparse distributions. In particular, [3] consider $s$-sparse distributions and establish minimax optimal rates under communication and privacy constraints, which are further improved by localization strategies in [14]. Complementary to minimax rates, [15] provides pointwise rates, governed by the half-norm of the distribution instead of its dimension. Our setting embraces heterogeneous data, and thus is a generalization of the one studied in above works.

**Robust estimation & learning.** Robust statistics and learning study algorithms resilient to unknown data corruption [30, 25]. The median-of-means method [32, 38, 39], partitions the data into subsets, computes an estimate from each, and takes their median. Similarly, some works study robustness from the optimization perspective, proposing to robustly aggregate gradients of the loss functions [48, 49, 9, 58]. We adapt some analysis techniques from [39, 58] to the significantly different setting of estimation with heterogeneity and communication constraints.

## 1.3 Notations

Throughout the paper, for an integer $d \geq 1$, we write $[d]$ for both $\{1, \ldots, d\}$ and $\{e_1, \ldots, e_d\} \subseteq \mathbb{R}^d$, where $e_k$ is the $k$-th canonical basis vector of $\mathbb{R}^d$. For a vector $\mathbf{v} \in \mathbb{R}^d$, we refer to the entries of $\mathbf{v}$ by both $[\mathbf{v}]_1, \ldots, [\mathbf{v}]_d$ and $v_1, \ldots, v_d$. We denote $\|\mathbf{v}\|_p = \left(\sum_{k \in [d]} |v_k|^p\right)^{\frac{1}{p}}$ for all $p > 0$ with $\|\mathbf{v}\|_0$ defined additionally as the number of non-zero entries. We let $\mathcal{P}_d := \{\mathbf{p} = (p_1, \ldots, p_d) \in [0, 1]^d : p_1 + \cdots + p_d = 1\}$ be the simplex of all $d$-dimensional discrete probability distributions. For $\mathbf{p} \in \mathcal{P}_d$, we denote by $\mathbb{B}_s(\mathbf{p})$ the $s$-distinct neighborhood $\{\mathbf{p}' \in \mathcal{P}_d : \|\mathbf{p}' - \mathbf{p}\|_0 \leq s\}$. For a random variable $X$, we denote $n$ i.i.d. copies of $X$ by $X^{[n]}$. Given any index set $\mathcal{I}$, we write $|\mathcal{I}|$ for its cardinality and denote by $[\mathbf{v}]_\mathcal{I}$ the sub-vector $([\mathbf{v}]_k)_{k \in \mathcal{I}}$ indexed by $\mathcal{I}$. We use the Bachmann-Landau asymptotic notations $\Omega(\cdot), \Theta(\cdot), O(\cdot)$ to hide constant factors, and use $\tilde{\Omega}(\cdot), \tilde{O}(\cdot)$ to also hide logarithmic factors. We denote the categorical distribution with class probability vector $\mathbf{p} \in \mathcal{P}_d$ by $\mathrm{Cat}(\mathbf{p})$. We use $\check{}$ and $\hat{}$ to indicate the intermediate estimate and the final estimate, respectively.

## 2 Problem Setup

We consider the problem of collaboratively learning distributions defined according to the following model of heterogeneity (see Figure 1 for an illustration). There are $T \geq 1$ clusters $\{\mathcal{C}^t \triangleq (X^{t,j})_{j \in [n]} :$

$t \in [T]\}$ of user datapoints, each of which contains $n$ i.i.d. local datapoints. Each datapoint $X^{t,j}$ is in a one-hot format, *i.e.*, $X^{t,j} \in \{e_1, e_2, \ldots, e_d\}$, and follows the categorical distribution $\mathrm{Cat}(\mathbf{p}^t)$ where $\mathbf{p}^t \in \mathcal{P}_d$ is unknown. Thus, user datapoints in the same cluster have an identical distribution $\mathbf{p}^t$, while the distribution $\mathbf{p}^t$ can vary, *i.e.*, be heterogeneous, across clusters $t \in [T]$. The datapoint $X^{t,j}$ is encoded by its user into a message $Y^{t,j}$, and then transmitted to a central server. We assume that the message sent by each datapoint is encoded into no more than $b$ bits and $b$ can be significantly smaller than $\log_2 d$ so that the communication is efficient. We also assume the server knows which cluster $t \in [T]$ each $Y^{t,j}$ belongs to, as well as the number of clusters $T$. This paper mainly addresses the communication bottleneck and does not involve privacy concerns.

The goal here is to collaboratively learn the distributions $\mathbf{p}^t$ from the collection of messages $\{Y^{t',[n]} \triangleq (Y^{t',j})_{j \in [n]} : t' \in [T]\}$ despite heterogeneity. More precisely, we aim to design per-cluster estimators $\widehat{\mathbf{p}}^t : \{Y^{t',[n]} : t' \in [T]\} \to \mathcal{P}_d$ to minimize the $\ell_2$ errors

$$\mathbb{E}[\|\widehat{\mathbf{p}}^t - \mathbf{p}^t\|_2^2], \quad \text{for all } t \in [T].$$

We also study the widely-used $\ell_1$ error metric (in addition to the $\ell_2$ metric). When $T = 1$, *i.e.*, all user datapoints are homogeneous and there is a single distribution to learn, the problem reduces to the one studied by [26, 6, 27, 2, 1].

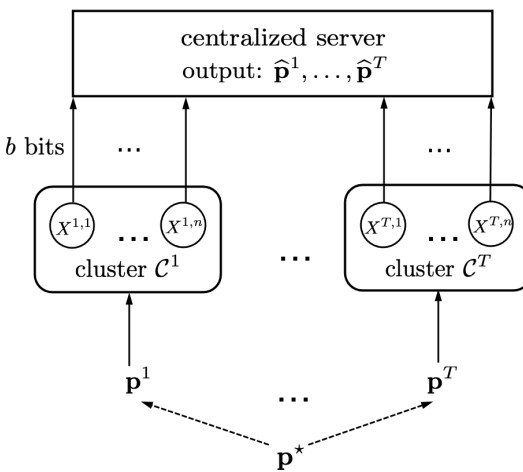

Figure 1: Learning distributions with heterogeneity and communication constraints.

**Model of heterogeneity.** In heterogeneous settings, collaboration among the users is most beneficial if the local distributions are related. We model this by assuming that the local distributions are sparse perturbations of an unknown *central distribution* $\mathbf{p}^\star \in \mathcal{P}_d$. The distribution $\mathbf{p}^t$ of each cluster $t$ differs from $\mathbf{p}^\star$ in at most $s \geq 0$ entries:

$$\|\mathbf{p}^t - \mathbf{p}^\star\|_0 \leq s, \quad \forall t \in [T]. \tag{1}$$

The central distribution $\mathbf{p}^\star$ can be viewed as the central structure across heterogeneous clusters of datapoints. The level of heterogeneity is controlled by the parameter $s$. When $s$ is much smaller than $d$, the local distributions differ from the center in a small number of entries.

While motivated by word frequencies of different texts, our model of sparse heterogeneity is also relevant for recommendation systems, where the high-dimensional item-preference vectors of users can vary sparsely [28, 42, 35, 7]; and medical risk scoring, where hospitals enjoy similar predicted results but with a few systematic differences in diagnosis behavior, healthcare utilization, etc [50, 43, 40].

## 3 Algorithm

We now introduce our method for leveraging heterogeneous data to improve per-cluster sample efficiency. We first discuss a hashing-based method to handle the communication constraint. Since communication between each user datapoint and the server is restricted to at most $b > 0$ bits (where we may have $2^b \ll d$), the datapoint $X^{t,j}$ needs to be encoded by an encoding function $W^{t,j} : \mathcal{X} \triangleq [d] \to \mathcal{Y}$. The $b$-bit constraint enforces $|\mathcal{Y}| \leq 2^b$. Then the encoded message $Y^{t,j} := W^{t,j}(X^{t,j})$ is sent to the server, where it is decoded and used.

Under relatively sophisticated protocols, the design of encoding functions can be *interactive* [15, 14, 1], *i.e.*, depend on previously sent messages. Here we adopt a *non-interactive* encoding-decoding scheme, based on uniform hashing [3, 14] where $W^{t,j}$ depends only on $X^{t,j}$ and is independent of other messages. Specifically, each datapoint $X^{t,j}$ is encoded via an independent random hash function $h^{t,j} : [d] \to [2^b]$. Upon receiving all messages, the server counts the empirical frequencies of all symbols, leading to hashed estimates $\breve{\mathbf{b}}^t$. The communication scheme based on uniform hashing

---

**Algorithm 1** SHIFT: Sparse Heterogeneity Inspired collaboration and Fine-Tuning

---

    **input:** individual hashed estimators $\breve{\mathbf{b}}^1, \ldots, \breve{\mathbf{b}}^T$, threshold parameter $\alpha$

    ▷ Stage I: Collaborative Learning

    Estimate $\mathbf{b}^\star$ via robust statistical methods: $\breve{\mathbf{b}}^\star \leftarrow \text{robust\_estimate}(\{\breve{\mathbf{b}}^t : t \in [T]\})$

    ▷ Stage II: Fine-Tuning

    **for** $k = 1, \ldots, d$ **do**

      **for** $t = 1, \ldots, T$ **do**

        $[\widehat{\mathbf{b}}^t]_k \leftarrow [\breve{\mathbf{b}}^\star]_k$ **if** $|[\breve{\mathbf{b}}^\star]_k - [\breve{\mathbf{b}}^t]_k| \le \sqrt{\alpha[\breve{\mathbf{b}}^t]_k/n}$, **else** $[\breve{\mathbf{b}}^t]_k$

        $[\widehat{\mathbf{p}}^t]_k \leftarrow \text{Proj}_{[0,1]}\left(\frac{2^b[\widehat{\mathbf{b}}^t]_k - 1}{2^b - 1}\right)$

      **end for**

    **end for**

    **output:** estimates $\widehat{\mathbf{p}}^1, \ldots, \widehat{\mathbf{p}}^T$

---

is summarized below.

(Encoding) : Send the message $Y^{t,j} = h^{t,j}(X^{t,j})$ encoded by a hash function $h^{t,j} : [d] \to [2^b]$;

(Decoding) : Count $N_k^t(Y^{t,[n]}) = |\{j \in [n] : h^{t,j}(k) = Y^{t,j}\}|$ and return $[\breve{\mathbf{b}}^t]_k = N_k^t/n$.

It is easy to verify that $\mathbb{E}[\breve{\mathbf{b}}^t] = [(2^b - 1)\mathbf{p}^t + 1]/2^b \triangleq \mathbf{b}^t$; and thus the hashed estimate $\breve{\mathbf{b}}^t$ is biased for $\mathbf{p}^t$. We also write $\mathbf{b}^\star = [(2^b - 1)\mathbf{p}^\star + 1]/2^b$ for the mean of a hashed datapoint sampled from the central distribution. More details on the hashed estimator $\breve{\mathbf{b}}^t$ are given in Appendix A.

## 3.1 The SHIFT Method

We now introduce the SHIFT method, which consists of two stages: *collaborative learning* and *fine-tuning*. The first stage estimates the central hashed distribution $\mathbf{b}^\star$ using all hashed estimates $\{\breve{\mathbf{b}}^t : t \in [T]\}$. This is achieved via tools from robust statistics such as the median or trimmed mean. The key insight here is that, since the heterogeneity is sparse, for each entry at which the individual distributions less mismatch with the central one, most datapoints (used to estimate that entry) are sampled from the probability of the central distribution. Hence, to estimate those entries of the central distribution, we can treat the datapoints generated by heterogeneous users as corrupted, and leverage robust statistical methods to mitigate their influences.

In the second stage—fine-tuning—we detect mismatched entries between individual hashed estimates $\breve{\mathbf{b}}^t$ and the central estimate $\breve{\mathbf{b}}^\star$. Recall that the central and individual distributions differ in only a few entries. For entries $k \in [d]$ such that $|[\breve{\mathbf{b}}^\star]_k - [\breve{\mathbf{b}}^t]_k|$ is below $(\alpha[\breve{\mathbf{b}}^\star]_k/n)^{1/2}$ for some threshold parameter $\alpha$, we may expect that $p_k^t = p_k^\star$. As a result, the estimate $[\breve{b}^\star]_k$ of $b_k^\star$ can be more accurate as it is learned collaboratively. Thus, we assign $[\breve{\mathbf{b}}^\star]_k$ as the final estimate $[\widehat{\mathbf{b}}^t]_k$ of $b_k^t$.

On the other hand, for the entries where the central and individual distributions differ, *i.e.*, $p_k^t \neq p_k^\star$, the threshold is more likely to be exceeded. In this case, we keep the individual estimate $[\breve{\mathbf{b}}^t]_k$ as $[\widehat{\mathbf{b}}^t]_k$. Finally, since the hashed distributions $\mathbf{b}^t$ are biased, we debias them in the final estimates of $\mathbf{p}^t$ where $\text{Proj}_{[0,1]}(\cdot)$ indicates truncating the input with upper/lower bound 1 and 0. Our method does not require sample splitting, despite using two stages, leading to increased sample-efficiency. We remark that the SHIFT does not require the knowledge of the sparse heterogeneity $s$.

**Knowledge transfer to new clusters.** The collaboratively learned central distribution from Algorithm 1 is adaptable to new clusters, with possibly few datapoints. This can be particularly beneficial for sample efficiency because most entries of the target distribution are well-estimated through collaborative learning. One can transfer those entries, and it suffices to estimate the few remaining entries, instead of the whole distribution. See Theorem 2 for the details. The knowledge transfer utility further motivates the importance of collaborative learning.

## 3.2 The Median-Based SHIFT

In this section, we provide statistical guarantees, in the form of upper bounds on the error, for the median-based SHIFT method, where $\text{robust\_estimate}(\{\breve{\mathbf{b}}^t : t \in [T]\})$ in Algorithm 1 is the entry-wise median. Specifically, we let

$$[\breve{\mathbf{b}}^\star]_k = \text{median}\big(\{[\breve{\mathbf{b}}^t]_k : t \in [T]\}\big), \quad \text{for each } k \in [d].$$

When there is no ambiguity, we write $\breve{\mathbf{b}}^\star = \text{median}\big(\{\breve{\mathbf{b}}^t\}_{t\in[T]}\big)$. We also provide results for the trimmed-mean-based SHIFT method, see Appendix D.

By setting the threshold parameter $\alpha$ in Algorithm 1 as $\alpha = \Theta(\ln(n))$, we prove the upper bounds on the final individual $\ell_2$ estimation errors as follows. The results for the $\ell_1$ error are in Appendix C.

**Theorem 1.** *Suppose $n \geq 2^{b+6}\ln(n)$ and $\alpha = \Theta(\ln(n))^2$. Then, for the median-based SHIFT method, for any $t \in [T]$,*

$$\mathbb{E}\left[\|\widehat{\mathbf{p}}^t - \mathbf{p}^t\|_2^2\right] = \tilde{O}\left(\frac{\max\{2^b, s\}}{2^b n} + \frac{d}{2^b T n} + \frac{d}{n^2}\right).$$

When $n \geq 2^{b+6}\ln(n)$ and $n = \Omega(2^b \min\{T, d/\max\{2^b, s\}\})$, the rate further becomes

$$\mathbb{E}\left[\|\widehat{\mathbf{p}}^t - \mathbf{p}^t\|_2^2\right] = \tilde{O}\left(\frac{\max\{2^b, s\}}{2^b n} + \frac{d}{2^b T n}\right). \tag{2}$$

The upper bound in (2) consists of two terms. The first term $\max\{2^b, s\}/(2^b n)$ is independent of the dimension $d$, and is a factor $d/\max\{2^b, s\}$ smaller compared to the rate $d/(2^b n)$ obtained by the minimax optimal method using only homogeneous datapoints [26]. Thus, it brings a significant benefit under sparse heterogeneity, *i.e.*, when $s \ll d$. Meanwhile, the second, dimension-dependent, term $d/(2^b T n)$ is $T$ times smaller than $d/(2^b n)$, since it depends on the *total sample-size* $Tn$ used collaboratively, despite heterogeneity. Therefore, our method shows a factor of $\min\{T, d/\max\{2^b, s\}\}$ improvement in sample efficiency, compared to previous work designed for homogeneous datapoints.

For completeness, we also consider a heuristic version of estimators of prior works [26, 15] in which all datapoints are pooled to learn a global distribution $T^{-1}\sum_{t\in[T]}\mathbf{p}^t$, which is then used by each cluster. While this uses all datapoints, it inevitably introduces a non-vanishing bias $\boldsymbol{\delta}^t = \mathbf{p}^t - T^{-1}\sum_{t'\in[T]}\mathbf{p}^{t'}$ in estimating individual distributions, and can behave poorly when the bias is large. See Table 1 for more details.

Finally, we discuss our results on knowledge transfer. The central estimator $\breve{\mathbf{b}}^\star$ is adaptable to a new cluster $\mathcal{C}^{T+1}$ in the following way. We adjust the fine-tuning procedure in Algorithm 1 to $[\widehat{\mathbf{b}}^{T+1}]_k \leftarrow [\breve{\mathbf{b}}^\star]_k$ if $|[\breve{\mathbf{b}}^\star]_k - [\breve{\mathbf{b}}^{T+1}]_k| \leq \sqrt{\alpha[\breve{\mathbf{b}}^{T+1}]_k/\tilde{n}}$, and $[\widehat{\mathbf{b}}^{T+1}]_k \leftarrow [\breve{\mathbf{b}}^{T+1}]_k$ otherwise, where $\tilde{n}$ is the size of $\mathcal{C}^{T+1}$. We then show the following result.

**Theorem 2.** *Let $\breve{\mathbf{b}}^{T+1}$ be the hashed estimate of any new cluster $\mathcal{C}^{T+1}$ with $\tilde{n}$ datapoints such that $n \geq \tilde{n} \geq 2^{b+6}\ln(\tilde{n})$ and $\tilde{n} = \Omega(2^b \min\{T, d/\max\{2^b, s\}\})$. Let the threshold parameter be $\alpha = \Theta(\ln(\tilde{n}))$. Then, the median-based SHIFT method has error bounded by*

$$\mathbb{E}\left[\|\widehat{\mathbf{p}}^{T+1} - \mathbf{p}^{T+1}\|_2^2\right] = \tilde{O}\left(\frac{\max\{2^b, s\}}{2^b \tilde{n}} + \frac{d}{2^b T n}\right).$$

Similarly, one can see that adaptation to new clusters with the median-based SHIFT method achieves a factor of $\min\{Tn/\tilde{n}, d/\max\{2^b, s\}\}$ improvement in sample-efficiency compared to the case where the distribution of the new cluster is estimated from scratch (*i.e.* without any knowledge transfer).

### 3.2.1 Highlights of Theoretical Analysis

In this section, we introduce the key analytical ideas behind the proof of Theorems 1 and 2. Our analysis is novel compared to previous analyses for methods with homogeneous datapoints. The

---

[2]To be precise, we require $\alpha = O(\ln(n))$ and $\alpha \geq c\ln(n)$ for some absolute constant $c$. The analogous statement applies in Theorem 2.

final individual estimation errors relate to the error of estimating the central hashed distribution $\mathbf{b}^\star$. However, we only expect high accuracy at the center for entries at which the heterogeneity is mild. To quantify the influence of heterogeneity, for any $0 < \eta \leq 1$, we define the set of $\eta$-*well-aligned entries* as

$$\mathcal{I}_\eta := \{k \in [d] : |\mathcal{B}_k| < \eta T\}, \quad \text{where} \quad \mathcal{B}_k \triangleq \{t \in [T] : b_k^t \neq b_k^\star, \text{ i.e., } p_k^t \neq p_k^\star\}$$

is the set of clusters whose distribution differs from $\mathbf{p}^\star$ in the $k$-th entry. We aim to estimate the $\eta$-well-aligned entries accurately by using robust statistical methods.

Further, we argue that there are few poorly-aligned entries, and they affect the final per-cluster error only mildly. By the pigeonhole principle, the number of entries that are not $\eta$-well-aligned is upper bounded by $|\mathcal{I}_\eta^c| \triangleq |[d] \backslash \mathcal{I}_\eta| \leq \frac{sT}{\eta T} = \frac{s}{\eta}$. Therefore, given an estimator $\breve{\mathbf{b}}^\star$ that is accurate for the $\eta$-well-aligned entries, the entries of $\mathbf{b}^\star$ can be estimated accurately except for at most $s/\eta$ entries. The following technical lemma bounds the error for each entry $k \in \mathcal{I}_\eta$.

**Lemma 1.** *Suppose $\breve{\mathbf{b}}^\star = \text{median}\big(\{\breve{\mathbf{b}}^t\}_{t \in [T]}\big)$. Then for any $0 < \eta \leq 1/5$ and $k \in \mathcal{I}_\eta$, it holds that*

$$\mathbb{E}[([\breve{\mathbf{b}}^\star]_k - [\mathbf{b}^\star]_k)^2] = \tilde{O}\left(\frac{|\mathcal{B}_k|^2 b_k^\star(1 - b_k^\star)}{T^2 n} + \frac{b_k^\star(1 - b_k^\star)}{Tn} + \frac{1}{n^2}\right).$$

Lemma 1 provides an upper bound relating to the frequency $|\mathcal{B}_k|/T$ of misalignment (smaller than $\eta$), and a variance term $b_k^\star(1 - b_k^\star)$. This result cannot be obtained by directly applying the standard Chernoff or Hoeffding bounds to random variables distributed in $[0, 1]$ as in previous works [15] for two reasons: 1) the datapoints are heterogeneous, 2) the variance $b_k^\star(1 - b_k^\star)$ here can be small, compared to general random variables in $[0, 1]$, implying more concentration than follows from Hoeffding's inequality. To address these issues, we analyze the concentration of the empirical $(1/2 \pm |\mathcal{B}_k|/T)$-quantiles to mitigate the influence of heterogeneity, and we also use Bernstein's inequality, which is variance-dependent [54], to obtain bounds relying on both the sample size $Tn$ and the variance $b_k^\star(1 - b_k^\star)$.

Also, the constant $1/5$, controlling the heterogeneity, is not essential (we choose $1/5$ for clarity). It can be replaced with any number below $1/2$ so that estimating the central probability distribution becomes possible, as the information conveyed by homogeneous datapoints dominates.

Lemma 1 reveals that well-aligned entries of the central distribution are accurately estimated. Thus one can use the central estimate for the entries where the central distribution $\mathbf{p}^t$ aligns with the target distribution $\mathbf{p}^t$. The remaining entries, that are neither well-aligned nor satisfy $p_k^\star = p_k^t$, can be estimated by the individual estimator. We argue that a properly chosen threshold parameter $\alpha$ filters out the desired entries to be estimated individually with high probability, leading to Theorems 1 and 2.

While estimating $\mathbf{p}^\star$ is not our main goal, one can readily obtain from Lemma 1 the following bound on estimating $\mathbf{p}^\star$ by summing up the errors for all entries $k \in [d] = \mathcal{I}_\eta$ with $\eta = \max_{k \in [d]} |\mathcal{B}_k|$. Corollary 1 reveals that the central distribution can be accurately estimated if the mismatching of distributions happens uniformly across all entries, *i.e.*, each entry differs in $O(sT/d)$ of clusters.

**Corollary 1.** *Let $\widehat{\mathbf{p}}^\star = \text{Proj}_{[0,1]}(\frac{2^b \breve{\mathbf{b}}^\star - 1}{2^b - 1})$ be obtained by the debiasing operation from Algorithm 1. Suppose $|\mathcal{B}_k| = O(sT/d)$ for any $k \in [d]$, with $\eta = \max_{k \in [d]} |\mathcal{B}_k|/T$, the median-based SHIFT method enjoys*

$$\mathbb{E}[\|\widehat{\mathbf{p}}^\star - \mathbf{p}^\star\|_2^2] = \tilde{O}\left(\frac{s^2}{d2^b n} + \frac{d}{2^b Tn} + \frac{d}{n^2}\right).$$

# 4 Lower Bounds

To complement our upper bounds, we now provide minimax lower bounds for estimating distributions under heterogeneity. Since our setting contains $T$ heterogeneous clusters of datapoints, our minimax error metric is slightly different from the one studied in [26, 6, 27, 2]. Using the $\ell_2$ error as the loss, the lower bound metric is defined as

$$\inf_{\substack{(W^{t',[n]})_{t' \in [T]} \\ \widehat{\mathbf{p}}^t}} \sup_{\substack{\mathbf{p}^\star \in \mathcal{P}_d \\ \{\mathbf{p}^{t'} : t' \in [T]\} \subseteq \mathbb{B}_s(\mathbf{p}^\star)}} \mathbb{E}\left[\|\widehat{\mathbf{p}}^t - \mathbf{p}^t\|_2^2\right], \tag{3}$$

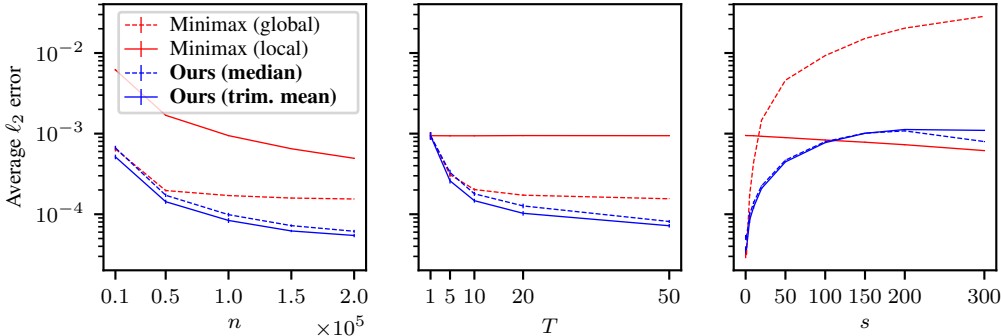

Figure 2: Average $\ell_2$ estimation error in synthetic experiment. (Left): Fixing $s = 5$, $T = 30$ and varying $n$. (Middle): Fixing $s = 5$, $n = 100,000$ and varying $T$. (Right): Fixing $T = 30$, $n = 100,000$ and varying $s$. The standard error bars are obtained from 10 independent runs.

where the supremum is taken over all possible central distributions $\mathbf{p}^\star \in \mathcal{P}_d$ and individual distributions $\{\mathbf{p}^t : t \in [T]\}$ in $\mathbb{B}_s(\mathbf{p}^\star) \triangleq \{\mathbf{p} \in \mathcal{P}_d : \|\mathbf{p} - \mathbf{p}^\star\|_0 \leq s\}$, and the infimum is taken over all estimation methods $\hat{\mathbf{p}}^t$ that use all heterogeneous messages $\{Y^{t',j} \triangleq W^{t',j}(X^{t',j}) : j \in [n], t' \in [T]\}$ encoded (possibly interactively) by any encoding functions $\{W^{t',j} : j \in [n], t' \in [T]\}$ with output in $[2^b]$, *e.g.*, the random hashing maps. The measure (3) characterizes the best possible worst-case performance of estimating distributions under our model of heterogeneity.

Since the supremum is taken over all distributions $\mathbf{p}^\star, \mathbf{p}^1, \ldots, \mathbf{p}^T$ in $\mathcal{P}_d$ such that $\|\mathbf{p}^t - \mathbf{p}^\star\|_0 \leq s$ for all $t \in [T]$, we consider two representative cases therein: 1) The *homogeneous* case where $\mathbf{p}^1 = \cdots = \mathbf{p}^T = \mathbf{p}^\star \in \mathcal{P}_d$. Then the setting essentially reduces to the single-cluster problem but with $nT$ datapoints, and the goal here is to estimate $\mathbf{p}^\star$, leading to the lower bound $\Omega(d/(2^bTn))$. 2) The $s/2$-*sparse* case where $\|\mathbf{p}^\star\|_0 \leq s/2$ and $\|\mathbf{p}^t\|_0 \leq s/2$ for all $t \in [T]$. Then it naturally holds that $\{\mathbf{p}^t : t \in [T]\} \subseteq \mathbb{B}_s(\mathbf{p}^\star)$. By constructing independent priors for $\{\mathbf{p}^t : t \in [T]\}$ and $\mathbf{p}^\star$, one can show that only datapoints generated by $\mathbf{p}^t$ itself are informative for estimating $\mathbf{p}^t$. In this case, we show the lower bound $\Omega(\max\{2^b, s\}/(2^bn))$. Combining the two cases, we find the following lower bound. The formal argument is provided in Appendix E.

**Theorem 3.** *For any—possibly interactive—estimation method, and for any $t \in [T]$, we have*

$$\inf_{\substack{(W^{t',[n]})_{t' \in [T]} \\ \hat{\mathbf{p}}^t}} \sup_{\substack{\mathbf{p}^\star \in \mathcal{P}_d \\ \{\mathbf{p}^{t'} : t' \in [T]\} \subseteq \mathbb{B}_s(\mathbf{p}^\star)}} \mathbb{E}[\|\hat{\mathbf{p}}^t - \mathbf{p}^t\|_2^2] = \Omega\left(\frac{\max\{2^b, s\}}{2^b n} + \frac{d}{2^b Tn}\right). \tag{4}$$

By a similar argument but with an additional $(T+1)$-st cluster of $\tilde{n}$ users, we obtain a lower bound for adapting to a new cluster.

**Theorem 4.** *For any—possibly interactive—estimation method, and a new cluster $\mathcal{C}^{T+1}$, we have*

$$\inf_{\substack{(W^{t',[n]})_{t' \in [T]} \\ W^{T+1,[\tilde{n}]}, \hat{\mathbf{p}}^{T+1}}} \sup_{\substack{\mathbf{p}^\star \in \mathcal{P}_d \\ \{\mathbf{p}^{t'} : t' \in [T+1]\} \subseteq \mathbb{B}_s(\mathbf{p}^\star)}} \mathbb{E}[\|\hat{\mathbf{p}}^{T+1} - \mathbf{p}^{T+1}\|_2^2] = \Omega\left(\frac{\max\{2^b, s\}}{2^b \tilde{n}} + \frac{d}{2^b Tn}\right). \tag{5}$$

Theorem 3 and 4, combined with the upper bounds in Section 3, imply that our method is minimax optimal up to logarithmic terms. We provide similar lower bounds for the $\ell_1$ error in Appendix E.

## 5   Experiments

We test SHIFT on synthetic data as well as the Shakespeare dataset [11]. As a baseline method, we use the estimator based on uniform grouping in [26] that is minimax optimal under homogeneity, *i.e.*, in the single-task regime. We apply the baseline method both locally and globally. In the local case, the estimator $\hat{\mathbf{p}}^t$ for each cluster is computed without datapoints from other clusters. In the global case, we pool data from all clusters, and compute estimators $\hat{\mathbf{p}} = \hat{\mathbf{p}}^1 = \cdots = \hat{\mathbf{p}}^T$. The performance measure for estimating $\hat{\mathbf{p}}^t$, $t \in [T]$ is taken as $T^{-1} \sum_{t=1}^T \|\mathbf{p}^t - \hat{\mathbf{p}}^t\|_2^2$.

| $k = 2$ | $b = 2$ | $b = 4$ | $b = 6$ | $b = 8$ |
|---|---|---|---|---|
| Unif. Group. (local) | $640 \pm 6.0$ | $142 \pm 1.2$ | $40 \pm 0.40$ | $14 \pm 0.13$ |
| Unif. Group. (global) | $33 \pm 1.8$ | $17 \pm 0.37$ | $14 \pm 0.081$ | $13 \pm 0.037$ |
| SHIFT (median) | $47 \pm 2.4$ | $21 \pm 0.66$ | $14 \pm 0.17$ | $11 \pm 0.10$ |
| SHIFT (trimmed mean) | $36 \pm 2.2$ | $19 \pm 0.51$ | $13 \pm 0.24$ | $10 \pm 0.062$ |
| $k = 3$ | $b = 2$ | $b = 4$ | $b = 6$ | $b = 8$ |
| Unif. Group. (local) | $15000 \pm 21$ | $3000 \pm 5.9$ | $720 \pm 2.1$ | $180 \pm 0.39$ |
| Unif. Group. (global) | $4400 \pm 5.7$ | $100 \pm 1.4$ | $38 \pm 0.35$ | $23 \pm 0.090$ |
| SHIFT (median) | $7300 \pm 9.6$ | $180 \pm 2.1$ | $53 \pm 1.0$ | $20 \pm 0.18$ |
| SHIFT (trimmed mean) | $5100 \pm 6.3$ | $140 \pm 2.3$ | $43 \pm 0.66$ | $18 \pm 0.18$ |

Table 2: Average $\ell_2$ error for estimating distributions of $k$-grams in the Shakespeare dataset. Numbers are scaled by $10^{-5}$.

### 5.1 Synthetic Data

We set the uniform distribution, $\mathbf{p}^\star = (1/d, \ldots, 1/d)$ as the central distribution. In Appendix F, we also experiment on the truncated geometric distribution and compare our method with the localization-refinement method [15]. Among the $d$ entries of $\mathbf{p}^\star$, we draw $s$ entries uniformly at random and assign new values for them uniformly at random over $[0, 1]$, with re-normalization to preserve their sum. We repeat this procedure $T$ times to obtain sparsely perturbed distributions $\mathbf{p}^1, \ldots, \mathbf{p}^T \in \mathcal{P}_d$. Then, $n$ i.i.d. datapoints $X^{t,1}, \ldots, X^{t,n} \sim \mathrm{Cat}(\mathbf{p}^t)$ are generated for each cluster $t \in [T]$. We set the dimension to $d = 300$ and run the simulation by varying $n, T, s$. As we see from (2), the error of our method depends on $s$ only when $2^b < s$. For this reason, we let $b = 2$ in our experiments.

We run SHIFT with the entry-wise median and entry-wise trimmed mean as the robust estimate. We set the threshold parameter $\alpha = \ln(n)$ and the trimming proportion $\omega = 0.1$. In Appendix F, we provide results for other choices of the hyperparameters $\alpha, \omega$ and discuss a heuristic for choosing $\alpha$. Figure 2 illustrates that our method outperforms the baselines for most choices of $n, T, s$. Specifically, as Theorem 1 predicts, the $\ell_2$ error of our method decreases as the number of clusters $T$ increases. On the other hand, when the baseline methods are applied globally, without considering heterogeneity, they show a bias that does not disappear as the sample size $n$ or the number of clusters $T$ increases. This shows that the fine-tuning step in SHIFT is crucial for the estimation of heterogeneous distributions. Finally, the right panel of Figure 2 shows that our method is effective only when $s$ is small compared to the dimension $d$; which highlights the crucial role of sparse heterogeneity. When $s$ is close to $d$, the distributions $\mathbf{p}^1, \ldots \mathbf{p}^T$ could be considerably different without any meaningful central structure, making collaboration less useful than local estimation.

### 5.2 Shakespeare Dataset

The Shakespeare dataset was proposed as a benchmark for federated learning in [11]. The dataset consists of dialogues of 1,129 speaking roles in Shakespeare's 35 different plays. In our experiment, we study the distribution of $k$-grams ($k$-tuples of consecutive letters from the 26-letter English alphabet, see Chapter 3 of [33]) appearing in the dialogues. We consider each play as a cluster $\mathcal{C}^t$ and estimate the distribution $\mathbf{p}^t \in \mathcal{P}_d$, $d = 26^k$ of $k$-grams. Since the ground-truth distribution $\mathbf{p}^t$ is unknown, we regard the empirical frequency as $\mathbf{p}^t$.

To verify the heterogeneity, we run the chi-squared goodness-of-fit test for each pair of distributions from distinct clusters $\mathbf{p}^u$ and $\mathbf{p}^v$. Resulting p-values were essentially zero within machine precision, which suggests that the distributions of $k$-grams are strongly heterogeneous. We also perform entry-wise tests comparing $[\mathbf{p}^u]_i$ and $[\mathbf{p}^v]_i$ for all $u \neq v \in [T], i \in [d]$. In total, 25.8% of the tests were rejected at the significance level of 5%. This again supports the heterogeneity.

We draw $n = 20,000$ datapoints with replacement from each cluster and test SHIFT with communication budgets $b \in \{2, 4, 6, 8\}$. We set the fine-tuning threshold $\alpha = \ln(n)$ and the trimming proportion $\omega = 0.1$, which we choose following the heuristic discussed in Appendix F. We repeat the experiment ten times by taking different datapoints and report the average $\ell_2$ error of estimation in Table 2. The standard deviations are small even over ten repetitions. SHIFT shows competitive performance on the empirical dataset, even though we do not rigorously know if the sparse heterogeneity model (1) applies.

# 6 Conclusion and Future Directions

We formulate the problem of learning distributions under sparse heterogeneity and communication constraints. We propose the SHIFT method, which first learns a central distribution, and then fine-tunes the estimate to adapt to individual distributions. We provide both theoretical and experimental results to show its sample-efficiency improvement compared to classical methods that target only homogeneous data. Many interesting directions remain to be explored, including investigating if there is a point-wise optimal method with rate depending on $\{\mathbf{p}^t : t \in [T]\}$ and $\mathbf{p}^\star$; and designing methods for other information constraints, such as local differential privacy constraints.

## Acknowledgements

During this work, Xinmeng Huang was supported in part by the NSF TRIPODS 1934960, NSF DMS 2046874 (CAREER), NSF CAREER award CIF-1943064; Donghwan Lee was supported in part by ARO W911NF-20-1-0080, DCIST, Air Force Office of Scientific Research Young Investigator Program (AFOSR-YIP) award #FA9550-20-1-0111.

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
