**Additional notations.** In the appendix, we use the following additional notations. For an integer $d \geq 1$, and a vector $\mathbf{v} \in \mathbb{R}^d$, the support $\mathrm{supp}(\mathbf{v}) = \{j \in [d] : \mathbf{v}_j \neq 0\}$ denotes the indices of non-zero entries. For an event $A$ on a probability space $(\Omega, B, P)$ (which is usually self-understood from the context), we denote by $I(A)$, $\mathbb{1}\{A\}$, or $\mathbb{1}(A)$ its indicator function, such that $I(A)(\omega) = 1$ if $\omega \in A$, and zero otherwise. We denote by $\Phi$ the cumulative distribution function of the standard normal random variable. For two scalars $a, b \in \mathbb{R}$, we write $a \wedge b = \min(a, b)$.

## A  Properties of Uniform Hashing

---

**Algorithm 2** Encoding-Decoding via Uniform Hashing

---

    **input:** cluster $\mathcal{C}^t$ with $n \geq 1$ users having data $X^{t,j}$, $j = 1, \ldots, n$
    **for** $j = 1, \ldots, n$ **do**
        Generate a uniformly random hash function $h^{t,j} : [d] \to [2^b]$ using shared randomness
        Encode message $Y^{t,j} = h^{t,j}(X^{t,j})$ and send it to the server         $\triangleright$ Encoding
    **end for**
    **for** $k = 1, \ldots, d$ **do**
        Count $N_k^t(Y^{t,[n]}) \leftarrow |\{j \in [n] : h^{t,j}(k) = Y^{t,j}\}|$         $\triangleright$ Decoding
        Estimate $\breve{b}_k^t \leftarrow N_k^t / n$
    **end for**
    **output:** $\widehat{\mathbf{b}}^t$

---

Recall that for all $t \in [T]$ and $k \in [d]$, $b_k^t = \frac{p_k^t(2^b - 1) + 1}{2^b} \in \left[\frac{1}{2^b}, 1\right]$.

**Proposition 1** (Properties of Hashed Estimates). *For each $t \in [T]$, suppose $\breve{\mathbf{b}}^t$ is computed in cluster $\mathcal{C}^t$ as in Algorithm 2 with i.i.d datapoints $X^{t,j} \sim \mathrm{Cat}(\mathbf{p}^t)$, $\forall j \in [n]$. Then, it holds that*

1. *$\breve{\mathbf{b}}^1, \ldots, \breve{\mathbf{b}}^T \in [0,1]$ are independent;*

2. *for any $t \in [T]$ and $k \in [d]$, $N_k^t \sim \mathrm{Binom}(n, b_k^t)$;*

3. *$\mathrm{supp}(\mathbf{p}^t - \mathbf{p}^\star) = \mathrm{supp}(\mathbf{b}^t - \mathbf{b}^\star)$ and $p_k^\star = 1$ (or 0) is equivalent to $b_k^\star = 1$ (or $\frac{1}{2^b}$, respectively).*

*Proof.* Property 1 holds because $\widehat{\mathbf{b}}^1, \ldots, \widehat{\mathbf{b}}^T$ are obtained by cluster-wise encoding-decoding of independent datapoints. To see property 2, we have for any $j \in [n]$ and $k \in [d]$ that

$$\mathbb{P}(h^{t,j}(k) = Y^{t,j}) = \mathbb{P}(k = X^{t,j}) + \mathbb{P}(k \neq X^{t,j} \text{ and } h^{t,j}(k) = h^{t,j}(X^{t,j}))$$

$$= p_k^t + (1 - p_k^t) \cdot \frac{1}{2^b} = b_k^t \in \left[\frac{1}{2^b}, 1\right].$$

Thus, $I(h^{t,j}(k) = Y^{t,j})$ is a Bernoulli variable with success probability $b_k^t$. Since each datapoint is encoded with an independent hash function, $N_k^t$ has a binomial distribution with $n$ trials and parameter $b_k^t$. Property 3 directly follows from $\mathbf{b}^t - \mathbf{b}^\star = (\mathbf{p}^t - \mathbf{p}^\star)(2^b - 1)/2^b$ and as $b > 0$. $\square$

**Proposition 2** (Property of Debiasing). *For any $\mathbf{y}, \mathbf{y}^\star \in \mathbb{R}^d$, let $\mathbf{x} = \mathrm{Proj}_{[0,1]}\left(\frac{2^b \mathbf{y} - 1}{2^b - 1}\right)$ and $\mathbf{x}^\star = \mathrm{Proj}_{[0,1]}\left(\frac{2^b \mathbf{y}^\star - 1}{2^b - 1}\right)$. Then it holds that for $q = 1, 2$, $\mathbb{E}[\|\mathbf{x} - \mathbf{x}^\star\|_q^q] = O(\mathbb{E}[\|\mathbf{y} - \mathbf{y}^\star\|_q^q])$. In particular, we have for $q = 1, 2$ and any $t \in [T]$, $\mathbb{E}[\|\widehat{\mathbf{p}}^t - \mathbf{p}^t\|_q^q] = O(\mathbb{E}[\|\widehat{\mathbf{b}}^t - \mathbf{b}^t\|_q^q])$, where $\widehat{\mathbf{p}}^t = \mathrm{Proj}_{[0,1]}\left(\frac{2^b \widehat{\mathbf{b}}^t - 1}{2^b - 1}\right)$ is the final per-cluster estimate obtained in Algorithm 1.*

*Proof.* Using the inequality that $|\mathrm{Proj}_{[0,1]}(x) - \mathrm{Proj}_{[0,1]}(y)| \leq |x - y|$ for any $x, y \in \mathbb{R}$, we have

$$\mathbb{E}[\|\mathbf{x} - \mathbf{x}^\star\|_q^q] = \sum_{k \in [d]} \mathbb{E}\left[\left|\mathrm{Proj}_{[0,1]}\left(\frac{2^b y_k - 1}{2^b - 1}\right) - \mathrm{Proj}_{[0,1]}\left(\frac{2^b y_k^\star - 1}{2^b - 1}\right)\right|^q\right]$$

$$\leq \sum_{k \in [d]} \mathbb{E}\left[\left|\frac{2^b(y_k - y_k^\star)}{2^b - 1}\right|^q\right] = \left(\frac{2^b}{2^b - 1}\right)^q \mathbb{E}[\|\mathbf{y} - \mathbf{y}^\star\|_q^q] = O(\mathbb{E}[\|\mathbf{y} - \mathbf{y}^\star\|_q^q]).$$

In the last step, we used that $2^b/(2^b - 1) \le 2$ for all $b \ge 1$, and thus the $O(\cdot)$ only depends on universal constants. $\qquad\square$

## B  General Lemmas

In this section, we state some general lemmas that will be used in the analysis.

**Lemma 2** (Berry-Esseen Theorem; [55]). *Assume that $Z_1, \ldots, Z_n$ are i.i.d. copies of a random variable $Z$ with mean $\mu$, variance $\sigma^2 > 0$, and such that $\mathbb{E}\left[|Z - \mu|^3\right] < \infty$. Then,*

$$\sup_{x \in \mathbb{R}} \left| \mathbb{P}\left\{ \sqrt{n} \frac{\bar{Z} - \mu}{\sigma} \le x \right\} - \Phi(x) \right| \le 0.4748 \frac{\gamma(Z)}{\sqrt{n}}.$$

*where $\bar{Z} = \frac{1}{n} \sum_{i=1}^n Z_i$ and $\gamma(Z) = \mathbb{E}[|Z - \mu|^3]/\sigma^3$ is the absolute skewness of $Z$.*

**Lemma 3** (Hoeffding's Inequality; [55]). *Let $Z_1, \ldots, Z_n \in [l, r]$, $l < r$, be independent random variables and let $\bar{Z} = \frac{1}{n} \sum_{j=1}^n Z_j$. Then for any $\delta \ge 0$,*

$$\max\{\mathbb{P}(\bar{Z} - \mathbb{E}[\bar{Z}] > \delta), \mathbb{P}(\bar{Z} - \mathbb{E}[\bar{Z}] < -\delta)\} \le \exp\left( -\frac{2n\delta^2}{(r - l)^2} \right).$$

**Lemma 4** (Bernstein's Inequality; [54]). *Let $Z_1, \ldots, Z_n$ be i.i.d. copies of a random variable $Z$ with $|Z - \mathbb{E}[Z]| \le M$, $M > 0$ and $\mathrm{Var}(Z_1) = \sigma^2 > 0$, and let $\bar{Z} = \frac{1}{n} \sum_{j=1}^n Z_j$. Then for any $\delta \ge 0$,*

$$\mathbb{P}(|\bar{Z} - \mathbb{E}[\bar{Z}]| > \delta) \le 2 \exp\left( -\frac{n\delta^2}{2(\sigma^2 + M\delta)} \right) \le 2 \exp\left( -\frac{n}{4} \min\left\{ \frac{\delta^2}{\sigma^2}, \frac{\delta}{M} \right\} \right). \tag{6}$$

The second inequality above directly follows from $\frac{1}{a+b} \ge \frac{1}{2} \min\{\frac{1}{a}, \frac{1}{b}\}$ for any $a, b > 0$. Note that (6) also allows $\sigma = 0$ because $\mathbb{P}(|\bar{Z} - \mathbb{E}[\bar{Z}]| > \delta) = 0$ and $\min\left\{ \delta^2/\sigma^2 \triangleq +\infty, \delta/M \right\} = \frac{\delta}{M}$. Therefore, we use this lemma for all $\sigma \ge 0$ below.

### B.1  Analysis Framework

For each $t \in [T]$, we denote by

$$\mathcal{K}_\alpha^t = \{k \in [d] : (\check{b}_k^\star - \check{b}_k^t)^2 \le \alpha \check{b}_k^t/n\} \tag{7}$$

the set of entries in which the central estimate $[\widehat{\mathbf{b}}^\star]_k$ is adapted to cluster $\mathcal{C}^t$. In this language, the final estimates can be expressed as $\widehat{b}_k^t = \check{b}_k^\star \mathbb{1}\{k \in \mathcal{K}_\alpha^t\} + \check{b}_k^t \mathbb{1}\{k \notin \mathcal{K}_\alpha^t\}$ for $t \in [T]$. Therefore, it holds that, for $q = 1, 2$,

$$\mathbb{E}[\|\widehat{\mathbf{b}}^t - \mathbf{b}^t\|_q^q] = \sum_{k \in [d]} \mathbb{E}[\mathbb{1}\{k \in \mathcal{K}_\alpha^t\}|\check{b}_k^\star - b_k^t|^q] + \sum_{k \in [d]} \mathbb{E}[\mathbb{1}\{k \notin \mathcal{K}_\alpha^t\}|\check{b}_k^t - b_k^t|^q]. \tag{8}$$

Let $\mathcal{I}^t \triangleq \{k \in [d] : b_k^t = b_k^\star$, i.e., $p_k^t = p_k^\star\}$ be set of entries at which the $t$-th cluster's distribution $\mathbf{p}^t$ *aligns* with the central distribution $\mathbf{p}^\star$. We next bound the two terms from (8) in Lemmas 5 and 6. These do not need the independence of $\check{\mathbf{b}}^\star$ and $\check{\mathbf{b}}^t$, and hence do not require sample splitting despite the division between stages.

**Lemma 5.** *For any $t \in [T]$, $\alpha \ge 1$ and $\eta \in (0, 1]$, with $\mathcal{K}_\alpha^t$ from (7), we have, for $q = 1, 2$*

$$\sum_{k \in [d]} \mathbb{E}[\mathbb{1}\{k \in \mathcal{K}_\alpha^t\}|\check{b}_k^\star - b_k^t|^q] = O\left( \mathbb{E}[\|\check{b}_{\mathcal{I}_\eta \cap \mathcal{I}^t}^\star - b_{\mathcal{I}_\eta \cap \mathcal{I}^t}^\star\|_q^q] + \sum_{k \notin \mathcal{I}_\eta \cap \mathcal{I}^t} \left( \frac{\alpha b_k^t}{n} \right)^{q/2} \right).$$

*Proof.* We first take $q = 1$. For any $k \in [d]$, clearly

$$\mathbb{E}[\mathbb{1}\{k \in \mathcal{K}_\alpha^t\}|\check{b}_k^\star - b_k^t|] \le \mathbb{E}[|\check{b}_k^\star - b_k^t|]. \tag{9}$$

We use this bound for $k \in \mathcal{I}_\eta \cap \mathcal{I}^t$. For $k \notin \mathcal{I}_\eta \cap \mathcal{I}^t$, we instead bound

$$\mathbb{E}[\mathbb{1}\{k \in \mathcal{K}_\alpha^t\}|\breve{b}_k^\star - b_k^t|] \leq \mathbb{E}[\mathbb{1}\{k \in \mathcal{K}_\alpha^t\}|\breve{b}_k^\star - \breve{b}_k^t|] + \mathbb{E}[\mathbb{1}\{k \in \mathcal{K}_\alpha^t\}|\breve{b}_k^t - b_k^t|].$$

If $k \in \mathcal{K}_\alpha^t$, it holds by definition that $|\breve{b}_k^\star - \breve{b}_k^t| \leq \sqrt{\alpha\breve{b}_k^t/n}$, thus we further have

$$\mathbb{E}[\mathbb{1}\{k \in \mathcal{K}_\alpha^t\}|\breve{b}_k^\star - b_k^t|] \leq \mathbb{E}\left[\mathbb{1}\{k \in \mathcal{K}_\alpha^t\}\sqrt{\alpha\breve{b}_k^t/n}\right] + \mathbb{E}[\mathbb{1}\{k \in \mathcal{K}_\alpha^t\}|\breve{b}_k^t - b_k^t|]$$

$$\leq \mathbb{E}\left[\sqrt{\alpha\breve{b}_k^t/n}\right] + \mathbb{E}[|\breve{b}_k^t - b_k^t|]. \tag{10}$$

By Jensen's inequality and since $n\breve{b}_k^t \sim \text{Binom}(n, b_k^t)$, we have

$$\mathbb{E}\left[\sqrt{\breve{b}_k^t}\right] \leq \sqrt{\mathbb{E}[\breve{b}_k^t]} = \sqrt{b_k^t} \tag{11}$$

and

$$\mathbb{E}[|\breve{b}_k^t - b_k^t|] \leq \sqrt{\mathbb{E}[(\breve{b}_k^t - b_k^t)^2]} = \sqrt{\frac{b_k^t(1 - b_k^t)}{n}} \leq \sqrt{\frac{b_k^t}{n}}. \tag{12}$$

Plugging (11) and (12) into (10), we find

$$\mathbb{E}[\mathbb{1}\{k \in \mathcal{K}_\alpha^t\}|\breve{b}_k^\star - b_k^t|] \leq (\sqrt{\alpha} + 1)\sqrt{\frac{b_k^t}{n}} = O\left(\sqrt{\frac{\alpha b_k^t}{n}}\right). \tag{13}$$

Summing up (9) over all entries in $\mathcal{I}_\eta \cap \mathcal{I}^t$ and summing up (13) over all entries not in $\mathcal{I}_\eta \cap \mathcal{I}^t$ leads to the claim for $q = 1$ in Lemma 5. The case $q = 2$ follows by a similar argument. □

**Lemma 6.** *For any $t \in [T]$, $\alpha \geq 1$ and $\eta \in (0, 1]$, with $\mathcal{K}_\alpha^t$ from (7), we have, for $q = 1, 2$*

$$\sum_{k \in [d]} \mathbb{E}[\mathbb{1}\{k \notin \mathcal{K}_\alpha^t\}|\breve{b}_k^t - b_k^t|^q]$$

$$= O\left(\sum_{k \in \mathcal{I}_\eta \cap \mathcal{I}^t} \mathbb{P}(k \notin \mathcal{K}_\alpha^t) \wedge \left(\frac{b_k^t(1 - b_k^t)}{n}\right)^{q/2} + \sum_{k \notin \mathcal{I}_\eta \cap \mathcal{I}^t} \left(\frac{b_k^t}{n}\right)^{q/2}\right).$$

*Proof.* For $q = 1$, note that

$$\mathbb{E}[\mathbb{1}\{k \notin \mathcal{K}_\alpha^t\}|\breve{b}_k^t - b_k^t|] \leq \mathbb{P}(k \notin \mathcal{K}_\alpha^t) \tag{14}$$

and

$$\mathbb{E}[\mathbb{1}\{k \notin \mathcal{K}_\alpha^t\}|\breve{b}_k^t - b_k^t|] \leq \mathbb{E}[|\breve{b}_k^t - b_k^t|]. \tag{15}$$

Combining (14), (15) with the first inequality in (12) for $k \in \mathcal{I}_\eta \cap \mathcal{I}^t$, and using the last inequality in (12) for $k \notin \mathcal{I}_\eta \cap \mathcal{I}^t$ leads to the claim with $q = 1$. We can similarly obtain the bound with $q = 2$. □

Combing Lemma 5 and 6 with (8), we find the following proposition:

**Proposition 3.** *For any $\alpha \geq 1$, and $q = 1, 2$, it holds that*

$$\mathbb{E}[\|\widehat{\mathbf{b}}^t - \mathbf{b}^t\|_q^q] = O\left(\sum_{k \notin \mathcal{I}_\eta \cap \mathcal{I}^t} \left(\frac{\alpha b_k^t}{n}\right)^{q/2}\right.$$

$$\left. + \sum_{k \in \mathcal{I}_\eta \cap \mathcal{I}^t} \mathbb{P}(k \notin \mathcal{K}_\alpha^t) \wedge \left(\frac{b_k^t(1 - b_k^t)}{n}\right)^{q/2} + \mathbb{E}[\|\breve{b}_{\mathcal{I}_\eta \cap \mathcal{I}^t}^\star - b_{\mathcal{I}_\eta \cap \mathcal{I}^t}^\star\|_q^q]\right).$$

Proposition 3 does not rely on how $\breve{\mathbf{b}}^\star$ is obtained. The next part is devoted to proving that when $\breve{\mathbf{b}}^\star$ is obtained via a certain robust estimate, the bounds in Proposition 3 are small for certain values of $\alpha$ and $\eta$.

# C  Median-Based Method

In this section, we provide the proofs for the median-based SHIFT method. We first re-state the detailed version of some key results that apply to both the $\ell_2$ and $\ell_1$ errors.

Below, we use $\sigma_k = \sqrt{b_k^\star(1 - b_k^\star)}$ to denote the standard deviation of the Bernoulli variable with success probability $b_k^\star = p_k^\star + (1 - p_k^\star)/2^b$. We also recall that $\mathcal{B}_k$ is defined as the set of clusters with distributions mismatched with the central distribution at the $k$-th entry, *i.e.*, $\mathcal{B}_k = \{t \in [T] : p_k^t \neq p_k^\star\}$, and $\mathcal{I}_\eta$ is defined as the $\eta$-well-aligned entries, *i.e.*, $\mathcal{I}_\eta = \{k \in [d] : |\mathcal{B}_k| < \eta T\}$.

**Lemma 7** (Detailed statement of Lemma 1). *Suppose* $\check{\mathbf{b}}^\star = \mathrm{median}\big(\{\check{\mathbf{b}}^t\}_{t \in [T]}\big)$. *Then for any* $0 < \eta \leq \frac{1}{5}$, $k \in \mathcal{I}_\eta$, *and* $q = 1, 2$, *it holds that*

$$\mathbb{E}[|\check{b}_k^\star - b_k^\star|^q] = \tilde{O}\left(\left(\frac{|\mathcal{B}_k|\sigma_k}{T\sqrt{n}}\right)^q + \left(\frac{\sigma_k}{\sqrt{Tn}}\right)^q + \left(\frac{1}{n}\right)^q\right).$$

Let us define, for $q = 1, 2$,

$$E(q) \triangleq E(q; n, d, b, T) := \frac{d}{(2^b Tn)^{q/2}} + \frac{d}{n^q}.$$

**Proposition 4.** *Suppose* $\check{\mathbf{b}}^\star = \mathrm{median}\big(\{\check{\mathbf{b}}^t\}_{t \in [T]}\big)$. *Then for any* $0 < \eta \leq \frac{1}{5}$ *and* $q = 1, 2$, *it holds that*

$$\mathbb{E}[\|\check{b}_{\mathcal{I}_\eta}^\star - b_{\mathcal{I}_\eta}^\star\|_q^q] = \tilde{O}\left(\sum_{k \in \mathcal{I}_\eta}\left(\frac{|\mathcal{B}_k|\sigma_k}{T\sqrt{n}}\right)^q + E(q)\right).$$

We omit the proofs of Proposition 4 and Theorem 6 (below), as Proposition 4 is a direct corollary of Lemma 7 by using $\sum_{k \in [d]} \sigma_k^q = O(d/2^{bq/2})$ for $q = 1, 2$, and Theorem 6 follows from the same analysis as Theorem 5.

**Theorem 5** (Detailed statement of Theorem 1). *Suppose* $n \geq 2^{b+6}\ln(n)$ *and* $\alpha \geq 2(8 + \sqrt{8\ln(n)})^2$ *with* $\alpha = O(\ln(n))$. *Then for the median-based SHIFT method, for any* $0 < \eta \leq \frac{1}{5}$, $q = 1, 2$, *and* $t \in [T]$,

$$\mathbb{E}\left[\|\hat{\mathbf{p}}^t - \mathbf{p}^t\|_q^q\right] = \tilde{O}\left(\sum_{k \notin \mathcal{I}_\eta \cap \mathcal{I}^t}\left(\frac{b_k^t}{n}\right)^{q/2} + \sum_{k \in \mathcal{I}_\eta \cap \mathcal{I}^t}\left(\frac{|\mathcal{B}_k|^2 b_k^\star}{T^2 n}\right)^{q/2} + E(q)\right).$$

*Furthermore, by setting* $\eta = \Theta(1)$ *with* $\eta \leq \frac{1}{5}$, *we have*

$$\mathbb{E}\left[\|\check{\mathbf{p}}^t - \mathbf{p}^t\|_q^q\right] = \tilde{O}\left(s^{1-q/2}\left(\frac{\max\{2^b, s\}}{2^b n}\right)^{q/2} + E(q).\right)$$

**Theorem 6** (Detailed statement of Theorem 2). *Suppose* $n \geq \tilde{n} \geq 2^{b+6}\ln(\tilde{n})$ *and* $\alpha \geq 2(8 + \sqrt{8\ln(\tilde{n})})^2$ *with* $\alpha = O(\ln(\tilde{n}))$. *Then the median-based SHIFT method for predicting the distribution of the new cluster with* $\tilde{n}$ *users achieves, for* $q = 1, 2$,

$$\mathbb{E}\left[\|\check{\mathbf{p}}^{T+1} - \mathbf{p}^{T+1}\|_q^q\right] = \tilde{O}\left(s^{1-q/2}\left(\frac{\max\{2^b, s\}}{2^b \tilde{n}}\right)^{q/2} + E(q).\right)$$

## C.1  Proof of Lemma 7

We first consider $T \leq 20\ln(n)$. In this case, by Bernstein's inequality (Lemma 4) with $M = 1$, we have for any $t \in [T]\backslash\mathcal{B}_k$ that for any $\delta \geq 0$,

$$\mathbb{P}\left(|\check{b}_k^t - b_k^\star| > \delta\right) \leq 2e^{-\frac{n}{4}\min\{\delta^2/\sigma_k^2, \delta\}}. \tag{16}$$

Taking $\delta = \max\{\sigma_k\sqrt{8\ln(n)/n}, 8\ln(n)/n\}$ in (16), we find

$$\mathbb{P}\left(|\breve{b}_k^t - b_k^\star| > \max\left\{\sigma_k\sqrt{\frac{8\ln(n)}{n}}, \frac{8\ln(n)}{n}\right\}\right) \leq \frac{2}{n^2}. \tag{17}$$

Since $|[T]\backslash\mathcal{B}_k| > \frac{T}{2}$ for any $k \in \mathcal{I}_\eta$ with $\eta \leq \frac{1}{5}$, we have, since $\breve{b}_k^\star = \text{median}\left(\{\breve{b}_k^t\}_{t\in[T]}\right)$, that there are $t_-, t_+ \in [T]\backslash\mathcal{B}_k$ with $\breve{b}_k^{t'} \leq \breve{b}_k^\star \leq \breve{b}_k^{t'}$. Hence, $|\breve{b}_k^\star - b_k^\star| \leq \max_{t\in[T]\backslash\mathcal{B}_k} |\breve{b}_k^t - b_k^\star|$.

Recall that for any random variable $0 \leq X \leq 1$ and any $\delta \geq 0$, $\mathbb{E}[X] \leq \delta + \mathbb{P}(X \geq \delta)$. Therefore, by taking the union bound of (17) over $k \in [T]\backslash\mathcal{B}_k$, and by the assumption that $T \leq 20\ln(n)$, we have

$$\mathbb{E}[|\breve{b}_k^\star - b_k^\star|] \leq \mathbb{E}[\max_{k\in[T]\backslash\mathcal{B}_k} |\breve{b}_k^t - b_k^\star|] \leq \sigma_k\sqrt{\frac{8\ln(n)}{n}} + \frac{8\ln(n)}{n} + \frac{2T}{n^2}$$

$$= O\left(\sigma_k\sqrt{\frac{\ln(n)}{n}} + \frac{\ln(n)}{n}\right) = O\left(\sigma_k\frac{\ln(n)}{\sqrt{Tn}} + \frac{\ln(n)}{n}\right) = \tilde{O}\left(\frac{\sigma_k}{\sqrt{Tn}} + \frac{1}{n}\right). \tag{18}$$

Similarly, we have

$$\mathbb{E}[(\breve{b}_k^\star - b_k^\star)^2] \leq \sigma_k^2\frac{8\ln(n)}{n} + \frac{64\ln(n)^2}{n^2} + \frac{2T}{n^2} = \tilde{O}\left(\frac{\sigma_k^2}{Tn} + \frac{1}{n^2}\right). \tag{19}$$

For each $k \in [d]$ with $b_k^\star \neq 1$ (recall that $b_k^\star \geq 1/2^b$ by definition), let $\gamma_k = (1 - 2b_k^\star(1 - b_k^\star))/\sqrt{b_k^\star(1 - b_k^\star)}$, and let $\tilde{F}_k(x) := \frac{1}{T-|\mathcal{B}_k|}\sum_{t\in[T]\backslash\mathcal{B}_k} \mathbb{1}(\breve{b}_k^t \leq x)$ be the empirical distribution function of $\{\breve{b}_k^t : b_k^t = b_k^\star\}$. Let $\varepsilon \in (0, 1/2)$ and $C_\varepsilon = \sqrt{2\pi}\exp((\Phi^{-1}(1-\varepsilon))^2/2)$. For $\delta \geq 0$, define, recalling $\eta T > |\mathcal{B}_k|$ for all $k \in \mathcal{I}_\eta$,

$$G_{k,T,\delta} = \frac{|\mathcal{B}_k|}{T} + \frac{10^{-8}}{Tn} + \sqrt{\frac{\delta}{T - |\mathcal{B}_k|}}$$

where the term $\frac{10^{-8}}{Tn}$ is used to overcome some challenges due to the discreteness of empirical distributions, and can be replaced with other suitably small terms (see the proof of Lemma 9). Further, define

$$G'_{k,T,\delta} = G_{k,T,\delta} + 0.4748\frac{\gamma_k}{\sqrt{n}}.$$

To prove Lemma 1 for $T > 20\ln(n)$, we need the following additional lemmas:

**Lemma 8.** *For any $\delta \geq 0$ such that*

$$G'_{k,T,\delta} \leq \frac{1}{2} - \varepsilon, \tag{20}$$

*it holds with probability at least $1 - 4e^{-2\delta}$ that*

$$\tilde{F}_k\left(b_k^\star + C_\varepsilon\frac{\sigma_k}{\sqrt{n}}G'_{k,T,\delta}\right) \geq \frac{1}{2} + \frac{|\mathcal{B}_k|}{T} + \frac{10^{-8}}{Tn}$$

*and*

$$\tilde{F}_k\left(b_k^\star - C_\varepsilon\frac{\sigma_k}{\sqrt{n}}G'_{k,T,\delta}\right) \leq \frac{1}{2} - \frac{|\mathcal{B}_k|}{T} - \frac{10^{-8}}{Tn}.$$

*Proof.* The proof essentially follows Lemma 1 of [58]. We provide the proof for the sake of being self-contained.

Let $Z_k^t = (\breve{b}_k^t - b_k^t)/\sqrt{\text{Var}(\breve{b}_k^t)}$ be a standardized version of $\breve{b}_k^t$ for each $t \in [T]$ and $k \in [d]$, with $b_k^\star \neq 1$. Let $\tilde{\Phi}_k(z) = \frac{1}{T-|\mathcal{B}_k|}\sum_{t\in[T]\backslash\mathcal{B}_k} \mathbb{1}(Z_k^t \leq z)$ be the empirical distribution of $\{Z_k^t : t \in [T]\backslash\mathcal{B}_k\}$. The distribution of $Z_k^t$ is identical $t \in [T]\backslash\mathcal{B}_k$, and we denote by $\Phi_k$ their common cdf.

By definition, $\mathbb{E}[\tilde{\Phi}_k(z)] = \Phi_k(z)$ for any $z \in \mathbb{R}$. Let $z_1 > 0 > z_2$ be such that $\Phi(z_1) = \frac{1}{2} + G'_{k,T,\delta}$ and $\Phi(z_2) = \frac{1}{2} - G'_{k,T,\delta}$, which exist due to (20). Then, by Lemma 2, we have

$$\Phi_k(z_1) \geq \frac{1}{2} + G_{k,T,\delta} \quad \text{and} \quad \Phi_k(z_2) \leq \frac{1}{2} - G_{k,T,\delta}. \tag{21}$$

Further, by the Hoeffding's inequality, we have for any $\delta \geq 0$ and $z \in \mathbb{R}$,

$$\left| \tilde{\Phi}_k(z) - \Phi_k(z) \right| \leq \sqrt{\frac{\delta}{T - |\mathcal{B}_k|}} \tag{22}$$

with probability at least $1 - 2e^{-2\delta}$. Then, by a union bound of (22) for $z = z_1, z_2$, and by (21), it holds with probability at least $1 - 4e^{-2\delta}$ that

$$\tilde{\Phi}_k(z_1) \geq \frac{1}{2} + \frac{|\mathcal{B}_k|}{T} + \frac{10^{-8}}{Tn} \quad \text{and} \quad \tilde{\Phi}_k(z_2) \leq \frac{1}{2} - \frac{|\mathcal{B}_k|}{T} - \frac{10^{-8}}{Tn}. \tag{23}$$

Finally, we bound the values of $z_1$ and $z_2$. By the mean value theorem, there exists $\xi \in [0, z_1]$ such that

$$G'_{k,T,\delta} = z_1 \Phi'(\xi) = \frac{z_1}{\sqrt{2\pi}} e^{-\frac{\xi^2}{2}} \geq \frac{z_1}{\sqrt{2\pi}} e^{-\frac{z_1^2}{2}}. \tag{24}$$

By (20) and the definition of $z_1$, we have $z_1 \leq \Phi^{-1}(1 - \varepsilon)$, and thus, by (24), we have

$$z_1 \leq \sqrt{2\pi} G'_{k,T,\delta} \exp\left( \frac{1}{2} (\Phi^{-1}(1 - \varepsilon))^2 \right). \tag{25}$$

Similarly, we have

$$z_1 \geq -\sqrt{2\pi} G'_{k,T,\delta} \exp\left( \frac{1}{2} (\Phi^{-1}(1 - \varepsilon))^2 \right). \tag{26}$$

Since for all $z$, $\tilde{\Phi}_k(z) = \tilde{F}_k(\sigma_k z / \sqrt{n} + b_k^\star)$, plugging (25) and (26) into (23), we find the conclusion of this lemma. $\qquad \square$

This leads to our next result.

**Lemma 9.** *For any $k \in [d]$ such that condition (20) holds, we have with probability at least $1 - 4e^{-2\delta}$ that*

$$\left| \check{b}_k^t - b_k^t \right| \leq C_\varepsilon \frac{\sigma_k}{\sqrt{n}} G'_{k,T,\delta} + \frac{0.4748 C_\varepsilon}{n}. \tag{27}$$

*Proof.* Let $\hat{F}_k$ be the empirical distribution function of $\{\check{b}_k^t : t \in [T]\}$, such that for all $x \in \mathbb{R}$, $\hat{F}_k(x) := \frac{1}{T} \sum_{t \in [T]} \mathbb{1}(\check{b}_k^t \leq x)$. We have

$$|\hat{F}_k(x) - \tilde{F}_k(x)| = \left| \frac{1}{T} \sum_{t \in [T]} \mathbb{1}(\check{b}_k^t \leq x) - \frac{1}{T - |\mathcal{B}_k|} \sum_{t \in [T] \setminus \mathcal{B}_k} \mathbb{1}(\check{b}_k^t \leq x) \right|$$

$$= \left| \frac{1}{T} \sum_{t \in \mathcal{B}_k} \mathbb{1}(\check{b}_k^t \leq x) - \frac{|\mathcal{B}_k|}{T(T - |\mathcal{B}_k|)} \sum_{t \in [T] \setminus \mathcal{B}_k} \mathbb{1}(\check{b}_k^t \leq x) \right|$$

$$\leq \max\left\{ \frac{1}{T} \cdot |\mathcal{B}_k|, \frac{|\mathcal{B}_k|}{T(T - |\mathcal{B}_k|)} \cdot (T - |\mathcal{B}_k|) \right\} = \frac{|\mathcal{B}_k|}{T}. \tag{28}$$

Define $\tilde{F}_k^-(x) := \frac{1}{T - |\mathcal{B}_k|} \sum_{t \in [T] \setminus \mathcal{B}_k} \mathbb{1}(\check{b}_k^t < x) \leq \tilde{F}_k(x)$. Then by (28) and Lemma 8, we have, with probability at least $1 - 4e^{-2\delta}$ that

$$\hat{F}_k\left( b_k^\star + C_\varepsilon \frac{\sigma_k}{\sqrt{n}} G'_{k,T,\delta} \right) \geq \frac{1}{2} + \frac{10^{-8}}{Tn} \quad \text{and} \quad \hat{F}_k^-\left( b_k^\star - C_\varepsilon \frac{\sigma_k}{\sqrt{n}} G'_{k,T,\delta} \right) \leq \frac{1}{2} - \frac{10^{-8}}{Tn}. \tag{29}$$

Let $\breve{b}_k^{(j)}$, $\forall j \in [T]$ be the $j$-th smallest element in $\{\breve{b}_k^t : t \in [T]\}$. Recalling the definition of the median, if $T$ is odd, then $\breve{b}_k^\star = \breve{b}_k^{((T+1)/2)}$. Therefore, $b_k^\star + C_\varepsilon \frac{\sigma_k}{\sqrt{n}} G'_{k,T,\delta} < \breve{b}_k^\star$ implies $\hat{F}_k\left(b_k^\star + C_\varepsilon \frac{\sigma_k}{\sqrt{n}} G'_{k,T,\delta}\right) \le \frac{1}{2} - \frac{1}{2T}$ and $b_k^\star - C_\varepsilon \frac{\sigma_k}{\sqrt{n}} G'_{k,T,\delta} > \breve{b}_k^\star$ implies $\hat{F}_k^-\left(b_k^\star - C_\varepsilon \frac{\sigma_k}{\sqrt{n}} G'_{k,T,\delta}\right) \ge \frac{1}{2} + \frac{1}{2T}$, leading to a contradiction with (29).

On the other hand, if $T$ is even, $\breve{b}_k^\star = (\breve{b}_k^{(T/2)} + \breve{b}_k^{(T/2+1)})/2$. Therefore, $b_k^\star + C_\varepsilon \frac{\sigma_k}{\sqrt{n}} G'_{k,T,\delta} < \breve{b}_k^\star$ implies $\hat{F}_k\left(b_k^\star + C_\varepsilon \frac{\sigma_k}{\sqrt{n}} G'_{k,T,\delta}\right) \le \frac{1}{2}$ and $b_k^\star - C_\varepsilon \frac{\sigma_k}{\sqrt{n}} G'_{k,T,\delta} > \breve{b}_k^\star$ implies $\hat{F}_k^-\left(b_k^\star - C_\varepsilon \frac{\sigma_k}{\sqrt{n}} G'_{k,T,\delta}\right) \ge \frac{1}{2}$, which is also contradictory to (29).

To summarize, it holds that
$$|\breve{b}_k^\star - b_k^\star| \le C_\varepsilon \frac{\sigma_k}{\sqrt{n}} G'_{k,T,\delta}$$
with probability at least $1 - 4e^{-2\delta}$. $\qquad\qquad\qquad\qquad\qquad\qquad\qquad\qquad\qquad\qquad\qquad$ $\square$

If $T \le 20\ln(n)$, Lemma 7 follows directly from (18) and (19). Now, given Lemma 8 and Lemma 9, we turn to prove Lemma 7 with $T \ge 20\ln(n)$. We first check condition (20). Since $|\mathcal{B}_k| \le \eta T$ for any $k \in \mathcal{I}_\eta$, $\eta \le \frac{1}{5}$, and $\gamma_k \sigma_k \le 1$, we have for each $k \in \mathcal{I}_\eta$ that
$$G'_{k,T,\delta} \le \eta + \frac{10^{-8}}{Tn} + \sqrt{\frac{5\delta}{4T}} + \frac{0.4748}{\sqrt{n}\sigma_k}.$$
When $T \ge 20\ln(n)$, for any $k \in [d]$ such that $\sigma_k \ge \frac{20}{\sqrt{n}(1-2\eta)}$, taking $\delta = \ln(n)$ above, we have
$$G'_{k,T,\delta} \le \eta + 10^{-8} + \frac{1}{4} + 0.4748\frac{1-2\eta}{20} \le \frac{1}{2} - 0.035755.$$
Therefore, condition (20) in Lemma 9 is satisfied with $\varepsilon = 0.035755$, for which we can check that $C_\varepsilon \le 13$. Thus, for any $\delta \le \ln(n)$,
$$\mathbb{P}\left(|\breve{b}_k^\star - b_k^\star| \ge 13\frac{\sigma_k}{\sqrt{n}} G_{k,T,\delta} + \frac{13}{n}\right) \le 4e^{-2\delta}. \tag{30}$$

Therefore, by (30), we have, using that for any random variable $0 \le X \le 1$ and any $0 \le r \le 1$, $\mathbb{E}[X] \le r + \mathbb{P}(X \ge r)$, and that for $\delta = (\ln n)/2$, one has $4e^{-2\delta} = 4/n$, we find
$$\mathbb{E}[|\breve{b}_k^\star - b_k^\star|] \le 13\frac{\sigma_k}{\sqrt{n}} G_{k,T,(\ln n)/2} + \frac{17}{n} = \tilde{O}\left(\frac{\sigma_k}{\sqrt{n}}\frac{|\mathcal{B}_k|}{T} + \frac{\sigma_k}{\sqrt{nT}} + \frac{1}{n}\right). \tag{31}$$
Similarly, by the Cauchy-Schwarz inequality, we also have
$$\mathbb{E}[(\breve{b}_k^\star - b_k^\star)^2] = O\left(\frac{\sigma_k^2}{n}\left(\frac{|\mathcal{B}_k|^2}{T^2} + \frac{\ln(n)}{T - |\mathcal{B}_k|}\right) + \frac{1}{n^2} + e^{-2\ln(n)}\right)$$
$$= \tilde{O}\left(\frac{\sigma_k^2}{n}\frac{|\mathcal{B}_k|^2}{T^2} + \frac{\sigma_k^2}{nT} + \frac{1}{n^2}\right). \tag{32}$$

On the other hand, for any $k \in [d] \backslash \mathcal{B}_k$ such that $\sigma_k < \frac{20}{\sqrt{n}(1-2\eta)}$, by Bernstein's inequality and a union bound, we have
$$\mathbb{P}\left(\max_{k \in [T]\backslash\mathcal{B}_k} |\breve{b}_k^t - b_k^\star| > \delta\right) \le 2(T - |\mathcal{B}_k|)e^{-\frac{n}{4}\min\{\delta^2/\sigma_k^2, \delta\}} \le 2Te^{-\frac{n}{4}\min\{\frac{n(1-2\eta)^2\delta^2}{400}, \delta\}}. \tag{33}$$

Since $|[T]\backslash\mathcal{B}_k| > \frac{T}{2}$, we have as before that $|\breve{b}_k^\star - b_k^\star| \le \max_{t \in [T]\backslash\mathcal{B}_k} |\breve{b}_k^t - b_k^\star|$. Taking $\delta = 4\max\{\ln(Tn^2), 10\sqrt{\ln(Tn^2)}\}/n$ in (33), with the same steps as above, we find
$$\mathbb{E}[|\breve{b}_k^\star - b_k^\star|] \le \mathbb{E}[\max_{k \in [T]\backslash\mathcal{B}_k} |\breve{b}_k^t - b_k^\star|] \le \delta + 2Te^{-\frac{n}{4}\min\{\frac{(1-2\eta)^2 n\delta^2}{400}, \delta\}}$$
$$\le \frac{4\max\{\ln(Tn^2), 10\sqrt{\ln(Tn^2)}\} + 2}{n} = \tilde{O}\left(\frac{1}{n}\right) \tag{34}$$

and

$$\mathbb{E}[(\check{b}_k^\star - b_k^\star)^2] \le \delta^2 + 2Te^{-\frac{n}{4}\min\{\frac{(1-2\eta)^2 n\delta^2}{400},\delta\}} = \tilde{O}\left(\frac{1}{n^2}\right). \tag{35}$$

To summarize, combining (31), (32) with (34),(35), we complete the proof when $T > 20\ln(n)$. Furthermore, by using $\sum_{k\in[d]}\sigma_k^q = O(d/2^{bq/2})$ for $q = 1, 2$, we directly reach Proposition 4.

## C.2    Proof of Theorem 5

We first consider the case where $T \le 20\ln(n)$. By definition, $\hat{b}_k^t$ is either equal to $\check{b}_k^t$ or $\check{b}_k^\star$, and the latter happens only when $k \in \mathcal{K}_\alpha^t$, i.e., $|\check{b}_k^\star - \check{b}_k^t| \le \sqrt{\alpha\check{b}_k^t/n}$. In this case, we have

$$|\hat{b}_k^t - b_k^t| = |\check{b}_k^\star - b_k^t| \le |\check{b}_k^t - b_k^t| + |\check{b}_k^\star - \check{b}_k^t| \le |\check{b}_k^t - b_k^t| + \sqrt{\frac{\alpha\check{b}_k^t}{n}}.$$

Therefore, we have $|\hat{b}_k^t - b_k^t| \le |\check{b}_k^t - b_k^t| + \sqrt{\alpha\check{b}_k^t/n}$ for all $k \in [d]$. This leads to

$$\mathbb{E}[\|\hat{\mathbf{b}}^t - \mathbf{b}^t\|_1] \le \mathbb{E}[\|\check{\mathbf{b}}^t - \mathbf{b}^t\|_1] + \sqrt{\frac{\alpha}{n}}\sum_{k\in[d]}\mathbb{E}\left[\sqrt{\check{b}_k^t}\right]$$

$$\le \mathbb{E}[\|\check{\mathbf{b}}^t - \mathbf{b}^t\|_1] + \sqrt{\frac{\alpha}{n}}\sum_{k\in[d]}\sqrt{\mathbb{E}[\check{b}_k^t]}, \tag{36}$$

where (36) holds by Jensen's inequality. By further using the Cauchy–Schwarz inequality, we have

$$\mathbb{E}[\|\check{\mathbf{b}}^t - \mathbf{b}^t\|_1] \le \sqrt{d\,\mathbb{E}[\|\check{\mathbf{b}}^t - \mathbf{b}^t\|_2^2]} = O\left(\frac{d}{\sqrt{2^b n}}\right) \tag{37}$$

and

$$\sum_{k\in[d]}\sqrt{\mathbb{E}[\check{b}_k^t]} = \sum_{k\in[d]}\sqrt{b_k^t} \le \sqrt{d\sum_{k\in[d]}b_k^t} = O\left(\frac{d}{\sqrt{2^b}}\right). \tag{38}$$

Plugging (37) and (38) into (36), we find

$$\mathbb{E}[\|\check{\mathbf{b}}^t - \mathbf{b}^t\|_1] = \tilde{O}\left(\frac{d}{\sqrt{2^b n}}\right) = \tilde{O}\left(\frac{d}{\sqrt{2^b Tn}}\right).$$

We can similarly prove

$$\mathbb{E}[\|\check{\mathbf{b}}^t - \mathbf{b}^t\|_2^2] = \tilde{O}\left(\frac{d}{2^b n}\right) = \tilde{O}\left(\frac{d}{2^b Tn}\right).$$

Next we prove the case where $T \ge 20\ln(n) = \Omega(\ln(n))$. We first consider the estimation errors over $k \in \mathcal{I}_\eta \cap \mathcal{I}^t$ such that $\sigma_k \ge \frac{20}{\sqrt{n}(1-2\eta)}$. Let $\mathcal{E}_k^t := \{\check{b}_k^t \ge \frac{1}{2}b_k^t \text{ and } |\check{b}_k^\star - b_k^\star| \le 8\sqrt{b_k^\star/n}\}$. If $n \ge 2^{b+6}\ln(n)$ and $0 < \eta \le 1/5$, then since $b_k^\star \ge \frac{1}{2^b}$ for any $k \in [d]$, we have

$$13\frac{\sigma_k}{\sqrt{n}}G_{k,T,\ln n} + \frac{13}{n} = 13\frac{\sigma_k}{\sqrt{n}}\left(\frac{|\mathcal{B}_k|}{T} + \frac{10^{-8}}{Tn} + \sqrt{\frac{\ln(n)}{T - |\mathcal{B}_k|}}\right) + \frac{13}{n}$$

$$\le 13\frac{\sigma_k}{\sqrt{n}}\left(\frac{|\mathcal{B}_k|}{T} + \frac{10^{-8}}{Tn} + \sqrt{\frac{5\ln(n)}{4T}}\right) + \frac{13}{n} \le 13\frac{\sigma_k}{\sqrt{n}}\left(\frac{1}{5} + 10^{-8} + \frac{1}{4}\right) + \frac{13}{\sqrt{n2^{b+6}\ln(n)}}$$

$$\le 13\frac{\sigma_k}{\sqrt{n}}\left(\frac{1}{5} + 10^{-8} + \frac{1}{4}\right) + \frac{13\sqrt{b_k^\star}}{\sqrt{n64\ln(n)}} \le 8\sqrt{\frac{b_k^\star}{n}}.$$

Hence, by (30), it holds that

$$\mathbb{P}\left(|\check{b}_k^\star - b_k^\star| \ge 8\sqrt{\frac{b_k^\star}{n}}\right) \le \frac{4}{n^2}. \tag{39}$$

By Bernstein's inequality and as $b_k^\star \geq \frac{1}{2^b}$, we have

$$\mathbb{P}\left(|\breve{b}_k^t - b_k^t| > \frac{b_k^t}{2}\right) \leq 2e^{-\frac{n}{4}\min\{\frac{b_k^t}{4(1-b_k^t)}, \frac{b_k^t}{2}\}} \leq 2e^{-\frac{nb_k^t}{16}} \leq 2e^{-\frac{n}{16 \cdot 2^b}} \leq \frac{2}{n^2}, \tag{40}$$

where the last inequality holds because $n \geq 2^{b+6}\ln(n)$. Combining (40) with (39), we find $\mathbb{P}((\mathcal{E}_k^t)^c) \leq \frac{6}{n^2}$. By definition, $k \notin \mathcal{K}_\alpha^t$ implies $|\breve{b}_k^\star - \breve{b}_k^t| > \sqrt{\alpha \breve{b}_k^t/n}$. On the event $\mathcal{E}_k^t$, this further implies $|\breve{b}_k^\star - \breve{b}_k^t| > \sqrt{\alpha b_k^t/2n}$. Combined with (39) and that $b_k^\star = b_k^t$ for any $k \in \mathcal{I}^t$, we have on the event $\mathcal{E}_k^t$

$$\left|\breve{b}_k^t - b_k^t\right| = \left|\breve{b}_k^t - b_k^\star\right| \geq \left|\breve{b}_k^t - \breve{b}_k^\star\right| - \left|\breve{b}_k^\star - b_k^\star\right| > \sqrt{\frac{b_k^t}{n}}\left(\sqrt{\frac{\alpha}{2}} - 8\right). \tag{41}$$

Let $\zeta \triangleq \sqrt{\alpha/2} - 8 \geq \sqrt{8\ln(n)}$ and $\mathcal{F}_k^t := \left\{\left|\widehat{b}_k^t - b_k^t\right| \geq \zeta\sqrt{b_k^t/n}\right\}$. By Bernstein's inequality, and using $n \geq 2^{b+6}\ln(n)$, we have

$$\mathbb{P}(\mathcal{F}_k^t) \leq 2e^{-\frac{n}{4}\min\{\frac{\zeta^2}{n(1-b_k^t)}, \zeta\sqrt{\frac{b_k^t}{n}}\}} \leq 2e^{-\min\{\frac{\zeta^2}{4}, \frac{\zeta}{4}\sqrt{\frac{n}{2^b}}\}} \leq \frac{2}{n^2}. \tag{42}$$

Combining (41) with (42), we find that for any $k \in \mathcal{I}_\eta \cap \mathcal{I}^t$ with $\sigma_k \geq \frac{20}{\sqrt{n}(1-2\eta)}$, it holds that

$$\mathbb{P}(k \notin \mathcal{K}_\alpha^t) \leq \mathbb{P}((\mathcal{E}_k^t)^c) + \mathbb{P}(\mathcal{E}_k^t \cap \{k \notin \mathcal{K}_\alpha^t\}) \leq \mathbb{P}((\mathcal{E}_k^t)^c) + \mathbb{P}(\mathcal{E}_k \cap \mathcal{F}_k^t)$$

$$\leq \mathbb{P}((\mathcal{E}_k^t)^c) + \mathbb{P}(\mathcal{F}_k^t) \leq \frac{8}{n^2}.$$

On the other hand for any $k \in \mathcal{I}_\eta \cap \mathcal{I}^t$ with $\sigma_k < \frac{20}{\sqrt{n}(1-2\eta)}$, we have

$$\sqrt{\frac{b_k^t(1 - b_k^t)}{n}} = \sqrt{\frac{b_k^\star(1 - b_k^\star)}{n}} = \frac{\sigma_k}{\sqrt{n}} = O\left(\frac{1}{n}\right).$$

Therefore, we have for all $k \in \mathcal{I}_\eta \cap \mathcal{I}^t$, and $q = 1, 2$

$$\min\left\{\mathbb{P}(k \notin \mathcal{K}_\alpha^t), \left(\frac{b_k^t(1 - b_k^t)}{n}\right)^{q/2}\right\} = O\left(\frac{1}{n^q}\right). \tag{43}$$

Since $\alpha = O(\ln(n))$, by (43) and Proposition 3, we obtain

$$\mathbb{E}[\|\widehat{\mathbf{b}}^t - \mathbf{b}^t\|_1] = \tilde{O}\left(\sum_{k \notin \mathcal{I}_\eta \cap \mathcal{I}^t}\sqrt{\frac{b_k^t}{n}} + \mathbb{E}[\|\breve{b}_{\mathcal{I}_\eta \cap \mathcal{I}^t}^\star - b_{\mathcal{I}_\eta \cap \mathcal{I}^t}^\star\|_1] + \frac{d}{n}\right). \tag{44}$$

Combining (44) with Proposition 4 and using that $\sigma_k \leq \sqrt{b_k^\star} = \sqrt{b_k^t}$ for any $k \in \mathcal{I}^t$, we have

$$\mathbb{E}[\|\widehat{\mathbf{b}}^t - \mathbf{b}^t\|_1] = \tilde{O}\left(\sum_{k \notin \mathcal{I}_\eta \cap \mathcal{I}^t}\sqrt{\frac{b_k^t}{n}} + \sum_{k \in \mathcal{I}_\eta \cap \mathcal{I}^t}\frac{|\mathcal{B}_k|}{T}\sqrt{\frac{b_k^t}{n}} + E(1)\right). \tag{45}$$

Since $|(\mathcal{I}^t)^c| = \|\mathbf{p}^t - \mathbf{p}^\star\|_0 \leq s$, by the Cauchy-Schwarz inequality, we have

$$\sum_{k \notin \mathcal{I}_\eta \cap \mathcal{I}^t}\sqrt{\frac{b_k^t}{n}} \leq \sum_{k \notin \mathcal{I}_\eta}\sqrt{\frac{b_k^t}{n}} + \sum_{k \notin \mathcal{I}^t}\sqrt{\frac{b_k^t}{n}} \leq \sum_{k \notin \mathcal{I}_\eta}\sqrt{\frac{b_k^t}{n}} + \sqrt{\frac{s\sum_{k \notin \mathcal{I}^t}b_k^t}{n}}$$

$$\leq \sum_{k \notin \mathcal{I}_\eta}\sqrt{\frac{b_k^t}{n}} + \sqrt{\frac{s\sum_{k \notin \mathcal{I}^t}((2^b - 1)p_k^t + 1)}{2^b n}} \leq \sum_{k \notin \mathcal{I}_\eta}\sqrt{\frac{b_k^t}{n}} + \sqrt{\frac{s(2^b - 1 + s)}{2^b n}}. \tag{46}$$

Plugging (46) into (44), we further obtain

$$\mathbb{E}[\|\widehat{\mathbf{b}}^t - \mathbf{b}^t\|_1] = \tilde{O}\left(\sum_{k \notin \mathcal{I}_\eta}\sqrt{\frac{b_k^t}{n}} + \sum_{k \in \mathcal{I}_\eta}\frac{|\mathcal{B}_k|}{T}\sqrt{\frac{b_k^t}{n}} + \sqrt{\frac{s\max\{2^b, s\}}{2^b n}} + E(1)\right). \tag{47}$$

Similarly, we can reach the following $\ell_2$ counterpart:

$$\mathbb{E}[\|\widehat{\mathbf{b}}^t - \mathbf{b}^t\|_2^2] = \tilde{O}\left(\sum_{k\notin\mathcal{I}_\eta}\frac{b_k^t}{n} + \sum_{k\in\mathcal{I}_\eta}\frac{|\mathcal{B}_k|^2}{T^2}\frac{b_k^t}{n} + \frac{\max\{2^b, s\}}{2^b n} + E(2)\right). \tag{48}$$

Note that $\sum_{k\in[d]}|\mathcal{B}_k|/T \le s$ and for any set $\mathcal{I}$ with $|\mathcal{I}| = \lceil\frac{s}{\eta}\rceil$,

$$\sum_{k\in\mathcal{I}}\sqrt{\frac{b_k^t}{n}} \le \sqrt{\frac{|\mathcal{I}|\sum_{k\in\mathcal{I}}((2^b-1)p_k^t+1)}{2^b n}} = O\left(\sqrt{\frac{s/\eta\max\{2^b, s/\eta\}}{2^b n}}\right).$$

Now, recalling the definition of $\mathcal{I}_\eta$, we apply Lemma 10 in (47) with $(r_k, x_k) = (\sqrt{b_k^t/n}, |\mathcal{B}_k|/T)$ for all $k\in[d]$, to find

$$\mathbb{E}[\|\widehat{\mathbf{b}}^t - \mathbf{b}^t\|_1] = \tilde{O}\left(\sqrt{\frac{s/\eta\max\{2^b, s/\eta\}}{2^b n}} + E(1)\right).$$

Therefore, for any $\eta = \Theta(1)$ with $\eta \le \frac{1}{5}$, we finally have

$$\mathbb{E}[\|\widehat{\mathbf{b}}^t - \mathbf{b}^t\|_1] = \tilde{O}\left(\sqrt{\frac{s\max\{2^b, s\}}{2^b n}} + E(1)\right).$$

Similarly, by combining (48) with Lemma 10, we have for any $\eta = \Theta(1)$ with $\eta \le \frac{1}{5}$,

$$\mathbb{E}[\|\widehat{\mathbf{b}}^t - \mathbf{b}^t\|_2^2] = \tilde{O}\left(\frac{\max\{2^b, s\}}{2^b n} + E(2)\right).$$

The result directly follows Proposition 2.

**Lemma 10.** *Given $\eta \in (0, 1]$, $r_k \ge 0$ for all $k \in [d]$, and for $q = 1, 2$, consider the functions $f_q : \{x \in \mathbb{R}^d : 0 \le x_k \le 1, \forall k \in [d]$ and $\sum_{k\in[d]} x_k \le s\} \to \mathbb{R}$, $f_q(x_1, \ldots, x_d) := \sum_{k\in[d]} r_k^q(\mathbb{1}\{x_k \ge \eta\} + x_k^q\mathbb{1}\{x_k < \eta\})$. Then it holds that*

$$\max_{x_1,\ldots,x_d} f_q(x_1\ldots,x_d) \le \sum_{k=1}^{\lceil s/\eta\rceil} r_{(k)}^q, \tag{49}$$

*where $r_{(1)} \ge \cdots \ge r_{(d)}$ is the non-decreasing rearrangement of $\{r_1, \ldots, r_d\}$.*

*Proof.* We only prove the result for $f_1$, and the result for function $f_2$ follows similarly. Note that $r_k(\mathbb{1}\{x_k \ge \eta\} + x_k\mathbb{1}\{r_k \ge \eta\})$ is increasing with respect to $r_k$ and $x_k$. To consider the maximum of the sum in $f$, by the rearrangement inequality, without loss of generality, we can assume $r_1 \ge r_2 \ge \cdots \ge r_d \ge 0$ and $1 \ge x_1 \ge x_2 \ge \cdots \ge x_d \ge 0$. In this case, we claim that the maximum is attained at $x_1 = \cdots = x_{\lfloor s/\eta\rfloor} = \eta$, $x_{\lfloor s/\eta\rfloor+1} = s - \eta\lfloor s/\eta\rfloor$, and $x_k = 0$ for all $k > \lfloor s/\eta\rfloor + 1$. Further, the maximum is $\sum_{k=1}^{\lfloor s/\eta\rfloor} r_k + r_{\lfloor s/\eta\rfloor+1}(s - \eta\lfloor s/\eta\rfloor)^2$, which is upper bounded by the right-hand side of (49). We now use the exchange argument to prove the claim.

Step 1: If there is some $k$ such that $x_k > \eta \ge x_{k+1}$, then defining $x'$ by letting $(x_k', x_{k+1}') = (\eta, x_k + x_{k+1} - \eta)$ while for other $j$, $x_j' = x_j$, increases the value of $f$. Therefore, the maximum is attained by $x$ such that for some $j$, $x_1 = \cdots = x_j = \eta > x_{j+1} \ge \cdots \ge x_d$.

Step 2: If there is some $k$ such that $\eta > x_k \ge x_{k+1} > 0$, then defining $x'$ by letting $(x_k', x_{k+1}') = (\min\{\eta, x_k + x_{k+1}\}, \max\{0, x_k + x_{k+1} - \eta\})$ while for other $j$, $x_j' = x_j$, increases the value of $f$. Therefore, combined with Step 1, the maximum is attained by $x$ such that for some $j$, $x_1 = \cdots = x_j = \eta > x_{j+1} \ge 0$ and $x_k = 0$ for all $k > j + 1$. Thus most one element lies in $(0, \eta)$.

Combining Step 1 and Step 2 above, we complete the proof of the claim, which further leads to (49). $\qquad\square$

# D  Trimmed-Mean-Based Method

In this section, we study the trimmed-mean-based estimator. Fix $\omega \in (0, 1/2)$. Specifically, for each $k \in [d]$, let $\mathcal{U}_k$ be the subset of $\{[\breve{\mathbf{p}}^t]_{t \in [T]}\}$ obtained by removing the largest and smallest $\omega T$ elements[3]. Then, the trimmed-mean-based method can be expressed as

$$\breve{b}_k^\star = \frac{1}{|\mathcal{U}_k|} \sum_{t \in \mathcal{U}_k} \breve{b}_k^t. \tag{50}$$

We also write $\breve{\mathbf{b}}^\star = \text{trmean}\big(\{\breve{\mathbf{b}}^t\}_{t \in [T]}, \omega\big)$. For any chosen trimming proportion $0 \leq \eta \leq \omega \leq \frac{1}{5}$, we control the estimation error of each $\eta$-well aligned entry. Intuitively, this is small because there are at most a fraction of $\eta$ elements from heterogeneous distributions. These are trimmed if they behave as outliers, and otherwise kept in $\mathcal{U}_k$. The error control for a single entry $k \in \mathcal{I}_\eta$ is in Lemma 11.

**Lemma 11.** *Suppose* $\breve{\mathbf{b}}^\star = \text{trmean}\big(\{\breve{\mathbf{b}}^t\}_{t \in [T]}, \omega\big)$ *such that* $0 \leq \omega \leq \frac{1}{5}$. *Then for each* $k \in \mathcal{I}_\eta$ *with* $0 < \eta \leq \omega$ *and any* $q = 1, 2$, *it holds that*

$$\mathbb{E}[|\breve{b}_k^\star - b_k^\star|^q] = \tilde{O}\left( \left(\omega^2 \frac{b_k^\star}{n}\right)^{q/2} + \left(\frac{b_k^\star}{Tn}\right)^{q/2} + \frac{1}{(Tn)^q} + \left(\frac{\omega}{n}\right)^q \right). \tag{51}$$

*Proof.* To prove Lemma 11, we need the following lemma.

**Lemma 12.** *For each* $k \in \mathcal{I}_\eta$ *with* $0 < \eta \leq \omega \leq \frac{1}{5}$, *and any* $\varepsilon_k, \delta_k \geq 0$, *it holds with probability at least* $1 - 2e^{-\frac{(T - |\mathcal{B}_k|)n}{4} \min\{\frac{\varepsilon_k^2}{\sigma_k^2}, \varepsilon_k\}} - 2(T - |\mathcal{B}_k|)e^{-\frac{n}{4} \min\{\frac{\delta_k^2}{\sigma_k^2}, \delta_k\}}$ *that*

$$|\breve{b}_k^\star - b_k^\star| \leq \frac{\varepsilon_k + 3\omega\delta_k}{1 - 2\omega}.$$

*Proof of Lemma 12.* By Bernstein's inequality and the union bound, we have for any $\varepsilon_k$, $\delta_k > 0$ that

$$\mathbb{P}\left( \left| \frac{1}{T - |\mathcal{B}_k|} \sum_{t \in [T] \setminus \mathcal{B}_k} \breve{b}_k^t - b_k^\star \right| > \varepsilon_k \right) \leq 2e^{-\frac{(T - |\mathcal{B}_k|)n}{4} \min\{\frac{\varepsilon_k^2}{\sigma_k^2}, \varepsilon_k\}}$$

and

$$\mathbb{P}\left( \max_{t \in [T] \setminus \mathcal{B}_k} |\breve{b}_k^t - b_k^\star| > \delta_k \right) \leq 2(T - |\mathcal{B}_k|)e^{-\frac{n}{4} \min\{\frac{\delta_k^2}{\sigma_k^2}, \delta_k\}}.$$

By the definition of $\breve{b}_k^\star$, we have

$$|\breve{b}_k^\star - b_k^\star| = \frac{1}{T(1 - 2\omega)} \left| \sum_{t \in \mathcal{U}_k} \breve{b}_k^t - b_k^\star \right|$$

$$= \frac{1}{T(1 - 2\omega)} \left| \sum_{t \in [T] \setminus \mathcal{B}_k} (\breve{b}_k^t - b_k^\star) - \sum_{t \in [T] \setminus (\mathcal{B}_k \cup \mathcal{U}_k)} (\breve{b}_k^t - b_k^\star) + \sum_{t \in \mathcal{B}_k \cap \mathcal{U}_k} (\breve{b}_k^t - b_k^\star) \right|$$

$$\leq \frac{1}{T(1 - 2\omega)} \left( \left| \sum_{t \in [T] \setminus \mathcal{B}_k} \breve{b}_k^t - b_k^\star \right| + \left| \sum_{i \notin \mathcal{B}_k \cup \mathcal{U}_k} \breve{b}_k^t - b_k^\star \right| + \left| \sum_{t \in \mathcal{B}_k \cap \mathcal{U}_k} \breve{b}_k^t - b_k^\star \right| \right).$$

It is clear that

$$\left| \sum_{t \in [T] \setminus (\mathcal{B}_k \cup \mathcal{U}_k)} (\breve{b}_k^t - b_k^\star) \right| \leq |[T] \setminus \mathcal{U}_k| \max_{t \in [T] \setminus \mathcal{B}_k} |\breve{b}_k^t - b_k^\star| = 2\omega T \max_{t \in [T] \setminus \mathcal{B}_k} |\breve{b}_k^t - b_k^\star|.$$

---

[3]To be precise, one can either trim $\lceil \omega T \rceil$ or $\lfloor \omega T \rfloor$ elements. From now on, we write $\omega T$ for conciseness without further notice.

Then we claim that $\left|\sum_{t\in\mathcal{B}_k\cap\mathcal{U}_k}\breve{b}_k^t - b_k^\star\right| \leq |\mathcal{B}_k|\max_{t\in[T]\setminus\mathcal{B}_k}|\breve{b}_k^t - b_k^\star|$. Let $\mathcal{Q}_{k,\mathrm{l}}$ and $\mathcal{Q}_{k,\mathrm{r}}$ be the indices of the trimmed elements on the left side and right side, respectively, *i.e.*, the smallest and largest $\omega T$ elements among $\{\breve{b}_k^t\}_{t\in[T]}$. If $\mathcal{B}_k\cap\mathcal{U}_k \neq \emptyset$, then $|\mathcal{U}_k\setminus\mathcal{B}_k| < T(1-2\omega)$. Furthermore, we have $|\mathcal{Q}_{k,\mathrm{l}}\cup(\mathcal{U}_k\setminus\mathcal{B}_k)| = |\mathcal{Q}_{k,\mathrm{r}}\cup(\mathcal{U}_k\setminus\mathcal{B}_k)| = \omega T + |\mathcal{U}_k\setminus\mathcal{B}_k| < T(1-\omega) \leq |[T]\setminus\mathcal{B}_k|$, which leads to $([T]\setminus\mathcal{B}_k)\cap\mathcal{Q}_{k,\mathrm{l}} \neq \emptyset$ and $(T\setminus\mathcal{B}_k)\cap\mathcal{Q}_{k,\mathrm{r}} \neq \emptyset$. In conclusion, we have $\max_{t\in\mathcal{U}_k}|\breve{b}_k^t - b_k^\star| \leq \max_{t\in[T]\setminus\mathcal{B}_k}|\breve{b}_k^t - b_k^\star|$, which completes the proof of the claim. Therefore, we have

$$|\breve{b}_k^\star - b_k^\star| \leq \frac{1}{T(1-2\omega)}\left(\left|\sum_{t\in[T]\setminus\mathcal{B}_k}|\breve{b}_k^t - b_k^\star|\right| + (2\omega T + |\mathcal{B}_k|)\max_{t\in[T]\setminus\mathcal{B}_k}|\breve{b}_k^t - b_k^\star|\right) \leq \frac{\varepsilon_k + 3\omega\delta_k}{1-2\omega}$$

with probability at least $1 - 2e^{-\frac{(T-|\mathcal{B}_k|)n}{4}\min\{\frac{\varepsilon_k^2}{\sigma_k^2},\varepsilon_k\}} - 2(T-|\mathcal{B}_k|)e^{-\frac{n}{4}\min\{\frac{\delta_k^2}{\sigma_k^2},\delta_k\}}$. $\qquad\square$

Given Lemma 12, by setting

$$\varepsilon_k = \max\left\{\frac{4\sigma_k\sqrt{\ln(T^2n^2)}}{\sqrt{(T-|\mathcal{B}_k|)n}}, \frac{8\ln(T^2n^2)}{(T-|\mathcal{B}_k|)n}\right\} = \tilde{O}\left(\frac{\sigma_k}{\sqrt{Tn}} + \frac{1}{Tn}\right)$$

and

$$\delta_k = \max\left\{\frac{4\sigma_k\sqrt{\ln(T^2(T-|\mathcal{B}_k|)n^2)}}{\sqrt{n}}, \frac{4\ln(T^2(T-|\mathcal{B}_k|)n^2)}{n}\right\} = \tilde{O}\left(\frac{\sigma_k}{\sqrt{n}} + \frac{1}{n}\right),$$

using that $1/(1-2\omega) \leq \frac{5}{3}$, and recalling $\sigma_k \leq \sqrt{b_k^\star}$, we have with probability at least $1 - \frac{4}{T^2n^2}$ that

$$
\begin{aligned}
|\breve{b}_k^\star - b_k^\star| &\leq \frac{\varepsilon_k + 3\omega\delta_k}{1-2\omega}\\
&\leq \frac{5\omega}{3}\max\left\{\frac{4\sqrt{b_k^\star\ln(T^3n^2)}}{\sqrt{n}}, \frac{4\ln(T^3n^2)}{n}\right\} + \frac{5}{3}\max\left\{\frac{4\sqrt{b_k^\star\ln(T^2n^2)}}{\sqrt{(T-|\mathcal{B}_k|)n}}, \frac{4\ln(T^2n^2)}{(T-|\mathcal{B}_k|)n}\right\} \quad (52)\\
&= \tilde{O}\left(\omega\sqrt{\frac{b_k^\star}{n}} + \frac{\omega}{n} + \frac{\sigma_k}{\sqrt{Tn}} + \frac{1}{Tn}\right),
\end{aligned}
$$

which implies

$$
\begin{aligned}
\mathbb{E}[|\breve{b}_k^\star - b_k^\star|] &= \tilde{O}\left(\omega\sqrt{\frac{b_k^\star}{n}} + \frac{\omega}{n} + \frac{\sigma_k}{\sqrt{Tn}} + \frac{1}{Tn} + \frac{1}{T^2n^2}\right)\\
&= \tilde{O}\left(\omega\sqrt{\frac{b_k^\star}{n}} + \sqrt{\frac{b_k^\star}{Tn}} + \frac{1}{Tn} + \frac{\omega}{n}\right).
\end{aligned}
$$

Similarly, we can obtain

$$
\begin{aligned}
\mathbb{E}[(\breve{b}_k^\star - b_k^\star)^2] &= \tilde{O}\left(\frac{\omega^2 b_k^\star}{n} + \frac{\omega^2}{n^2} + \frac{\sigma_k^2}{Tn} + \frac{1}{T^2n^2} + \frac{1}{T^2n^2}\right)\\
&= \tilde{O}\left(\omega^2\frac{b_k^\star}{n} + \frac{b_k^\star}{Tn} + \frac{1}{T^2n^2} + \frac{\omega^2}{n^2}\right).
\end{aligned}
$$

$\qquad\square$

Given these results, we readily establish the following bound on the total error over all $\eta$-well-aligned entries.

**Proposition 5.** *Suppose $\breve{\mathbf{b}}^\star = \mathrm{trmean}\big(\{\breve{\mathbf{b}}^t\}_{t\in[T]}, \omega\big)$ such that $0 \leq \omega \leq 1/5$. Then for each $k \in \mathcal{I}_\eta$ with $0 < \eta \leq \omega$ and any $q = 1, 2$, it holds that*

$$\mathbb{E}[\|\breve{b}_{\mathcal{I}_\eta}^\star - b_{\mathcal{I}_\eta}^\star\|_q^q] = \tilde{O}\left(d\left(\frac{\omega^2}{2^b n}\right)^{q/2} + \frac{d}{(2^b Tn)^{q/2}} + \frac{d}{(Tn)^q} + d\left(\frac{\omega}{n}\right)^q\right).$$

By setting $\alpha = \Theta(\ln(Tn))$, we find the following result.

**Theorem 7.** *Suppose $n \geq 2^{b+5} \ln(Tn)$ and $\alpha \geq 2(8 + \sqrt{8 \ln(Tn)})^2$ with $\alpha = O(\ln(Tn))$. Then for the trimmed-mean-based SHIFT method, for any $0 < \omega \leq \frac{1}{5}$, $t \in [T]$ and $q = 1, 2$,*

$$\mathbb{E}\left[\|\widehat{\mathbf{p}}^t - \mathbf{p}^t\|_q^q\right] = \tilde{O}\left(\left(\frac{s}{\omega}\right)^{1-q/2}\left(\frac{\max\{2^b, s/\omega\}}{2^b n}\right)^{q/2} + d\left(\frac{\omega^2}{2^b n}\right)^{q/2} + \frac{d}{(2^b Tn)^{q/2}}\right).$$

*Proof.* To apply Proposition 3, we need to bound $\sum_{k \in \mathcal{I}_\eta \cap \mathcal{I}^t} \min\{\mathbb{P}(k \notin \mathcal{K}_\alpha^t), \sqrt{b_k^t(1 - b_k^t)/n}\}$ and $\sum_{k \in \mathcal{I}_\eta \cap \mathcal{I}^t} \min\{\mathbb{P}(k \notin \mathcal{K}_\alpha^t), b_k^t(1 - b_k^t)/n\}$.

Let $\mathcal{E}_k^t := \{\breve{b}_k^t \geq \frac{1}{2} b_k^t$ and $|\breve{b}_k^\star - b_k^\star| \leq 8\sqrt{b_k^\star \ln(T^3 n^2)/n}\}$. For each entry $k \in \mathcal{I}_\eta \cap \mathcal{I}^t$, since $n \geq 2^b \ln(T^3 n^2)$ and $b_k^\star \leq \frac{1}{2^b}$, we have $\frac{1}{n} \leq \sqrt{\frac{b_k^\star}{n \ln(T^3 n^2)}}$. By (52), we have with probability at least $1 - \frac{4}{T^2 n^2}$ that

$$|\breve{b}_k^t - b_k^t| \leq \frac{5\omega}{3} \max\left\{\frac{4\sqrt{b_k^\star \ln(T^3 n^2)}}{\sqrt{n}}, \frac{4 \ln(T^3 n^2)}{n}\right\} + \frac{5}{3} \max\left\{\frac{4\sqrt{b_k^\star \ln(T^2 n^2)}}{\sqrt{(T - |\mathcal{B}_k|)n}}, \frac{4 \ln(T^2 n^2)}{(T - |\mathcal{B}_k|)n}\right\}$$

$$\leq \frac{4}{3}\sqrt{\frac{b_k^\star \ln(T^3 n^2)}{n}} + \frac{20}{3}\sqrt{\frac{b_k^\star \ln(T^2 n^2)}{(T - |\mathcal{B}_k|)n}} \leq 8\sqrt{\frac{b_k^\star \ln(T^3 n^2)}{n}}. \tag{53}$$

By Bernstein's inequality and as $b_k^\star \geq \frac{1}{2^b}$, we have

$$\mathbb{P}\left(|\breve{b}_k^t - b_k^t| > \frac{b_k^t}{2}\right) \leq 2e^{-\frac{n}{4} \min\{\frac{b_k^t}{4(1-b_k^t)}, \frac{b_k^t}{2}\}} \leq 2e^{-\frac{n b_k^t}{16}} \leq 2e^{-\frac{n}{16 \cdot 2^b}} \leq \frac{2}{T^2 n^2}, \tag{54}$$

where the last inequality is because $n \geq 2^{b+5} \ln(Tn)$. Combining (53) with (54), we find $\mathbb{P}((\mathcal{E}_k^t)^c) \leq \frac{6}{T^2 n^2}$. Now following the argument from (41)-(43), we can obtain that for all $k \in \mathcal{I}_\eta \cap \mathcal{I}^t$,

$$\mathbb{P}(k \notin \mathcal{K}_\alpha^t) = O\left(\frac{1}{T^2 n^2}\right).$$

Since $\alpha = O(\ln(Tn))$, by applying (43) to Proposition 3 with $\eta = \omega$ and using Proposition 5 with $n = \Omega(2^b)$, we find

$$\mathbb{E}[\|\widehat{\mathbf{b}}^t - \mathbf{b}^t\|_1] = \tilde{O}\left(\sum_{k \notin \mathcal{I}_\omega \cap \mathcal{I}^t} \sqrt{\frac{b_k^t}{n}} + \frac{d\omega}{\sqrt{2^b n}} + \frac{d}{\sqrt{2^b Tn}}\right) \tag{55}$$

and

$$\mathbb{E}[\|\widehat{\mathbf{b}}^t - \mathbf{b}^t\|_2^2] = \tilde{O}\left(\sum_{k \notin \mathcal{I}_\omega \cap \mathcal{I}^t} \frac{b_k^t}{n} + \frac{d\omega}{2^b n} + \frac{d}{2^b Tn}\right).$$

Note that $|(\mathcal{I}_\omega \cap \mathcal{I}^t)^c| \leq |\mathcal{I}_\omega^c| + |(\mathcal{I}^t)^c| \leq s/\omega + s = O(s/\omega)$ and

$$\sum_{k \notin \mathcal{I}_\omega \cap \mathcal{I}^t} \sqrt{\frac{b_k^t}{n}} \leq \sqrt{|(\mathcal{I}_\omega \cap \mathcal{I}^t)^c| \sum_{k \notin \mathcal{I}_\omega \cap \mathcal{I}^t} \frac{b_k^t}{n}} = \sqrt{\frac{|(\mathcal{I}_\omega \cap \mathcal{I}^t)^c| \max\{2^b, |(\mathcal{I}_\omega \cap \mathcal{I}^t)^c|\}}{2^b n}}$$

$$= \sqrt{\frac{s/\omega \max\{2^b, s/\omega\}}{2^b n}}. \tag{56}$$

Plugging (56) into (55) and using $\mathbb{E}[\|\widehat{\mathbf{p}}^t - \mathbf{p}^t\|_1] = O(\mathbb{E}[\|\widehat{\mathbf{b}}^t - \mathbf{b}^t\|_1])$, we find the conclusion in terms of the $\ell_1$ error. The results in terms of the $\ell_2$ error can be obtained similarly. $\qquad \square$

# E Lower Bounds

In this section, we provide the proofs for the minimax lower bounds for estimating distributions under our heterogeneity model. We first re-state the detailed version the lower bounds that apply to both the $\ell_2$ and $\ell_1$ errors.

**Theorem 8** (Detailed statement of Theorem 3). *For any—possibly interactive—estimation method, and for any $t \in [T]$ and $q = 1, 2$, we have*

$$\inf_{\substack{(W^{t',[n]})_{t' \in [T]} \\ \widehat{\mathbf{p}}^t}} \sup_{\substack{\mathbf{p}^\star \in \mathcal{P}_d \\ \{\mathbf{p}^{t'}:t' \in [T]\} \subseteq \mathbb{B}_s(\mathbf{p}^\star)}} \mathbb{E}[\|\widehat{\mathbf{p}}^t - \mathbf{p}^t\|_q^q] = \Omega \left( s^{1-q/2} \left( \frac{\max\{2^b, s\}}{2^b n} \right)^{q/2} + \frac{d}{(2^b T n)^{q/2}} \right).$$

$$(57)$$

**Theorem 9** (Detailed statement of Theorem. 4). *For any—possibly interactive—estimation method, and a new cluster $\mathcal{C}^{T+1}$, we have*

$$\inf_{\substack{(W^{t',[n]})_{t' \in [T]} \\ W^{T+1,[\tilde{n}]},\widehat{\mathbf{p}}^{T+1}}} \sup_{\substack{\mathbf{p}^\star \in \mathcal{P}_d \\ \{\mathbf{p}^{t'}:t' \in [T+1]\} \subseteq \mathbb{B}_s(\mathbf{p}^\star)}} \mathbb{E}[\|\widehat{\mathbf{p}}^{T+1} - \mathbf{p}^{T+1}\|_q^q]$$

$$= \Omega \left( s^{1-q/2} \left( \frac{\max\{2^b, s\}}{2^b \tilde{n}} \right)^{q/2} + \frac{d}{(2^b T n)^{q/2}} \right).$$

We omit the proof of Theorem 9 since it follows from the same analysis as Theorem 8.

## E.1 Proof of Theorem 8

As discussed in Section 4, we will prove (57) by considering two special cases of our sparse heterogeneity model:

1. The *homogeneous* case where $\mathbf{p}^1 = \cdots = \mathbf{p}^T = \mathbf{p}^\star \in \mathcal{P}_d$.
2. The *$s/2$-sparse* case where $\|\mathbf{p}^\star\|_0 \leq s/2$ and $\|\mathbf{p}^t\|_0 \leq s/2$ for all $t \in [T]$.

Therefore, it naturally holds that

$$\inf_{\substack{(W^{t',[n]})_{t' \in [T]} \\ \widehat{\mathbf{p}}^t}} \sup_{\substack{\mathbf{p}^\star \in \mathcal{P}_d \\ \{\mathbf{p}^{t'}:t' \in [T]\} \subseteq \mathbb{B}_s(\mathbf{p}^\star)}} \mathbb{E}[\|\widehat{\mathbf{p}}^t - \mathbf{p}^t\|_q^q] \geq \inf_{\substack{(W^{t,[n]})_{t \in [T]} \\ \widehat{\mathbf{p}}^\star}} \sup_{\mathbf{p}^\star \in \mathcal{P}_d} \mathbb{E}[\|\widehat{\mathbf{p}}^\star - \mathbf{p}^\star\|_q^q] \qquad (58)$$

and

$$\inf_{\substack{(W^{t',[n]})_{t' \in [T]} \\ \widehat{\mathbf{p}}^t}} \sup_{\substack{\mathbf{p}^\star \in \mathcal{P}_d \\ \{\mathbf{p}^{t'}:t' \in [T]\} \subseteq \mathbb{B}_s(\mathbf{p}^\star)}} \mathbb{E}[\|\widehat{\mathbf{p}}^t - \mathbf{p}^t\|_q^q] \geq \inf_{\substack{(W^{t',[n]})_{t' \in [T]} \\ \widehat{\mathbf{p}}^t}} \sup_{\substack{\|\mathbf{p}^{t'}\|_0 \leq s/2 \\ \forall\, t' \in [T]}} \mathbb{E}[\|\widehat{\mathbf{p}}^t - \mathbf{p}^t\|_q^q]. \qquad (59)$$

For the first case, combining (58) with the existing lower bound result [6, Cor 7] and [26, Thm 2] for the homogeneous setup, where all datapoints are generated by a single distribution, that for any estimation method (possibly based on interactive encoding),

$$\inf_{\substack{(W^{t,[n]})_{t \in [T]} \\ \widehat{\mathbf{p}}^\star}} \sup_{\mathbf{p}^\star \in \mathcal{P}_d} \mathbb{E}[\|\widehat{\mathbf{p}}^\star - \mathbf{p}^\star\|_q^q] = \Omega \left( \frac{d}{(2^b T n)^{q/2}} \right),$$

we prove that the lower bound is at least of the order of the second term in (57).

For the second case, without loss of generality, we assume $s$ is even. This can be achieved by considering $s - 1$ instead of $s$, if necessary. Recall that $\text{supp}(\cdot)$ denotes the indices of non-zero entries of a vector. Fixing any $t \in [T]$, we further consider the scenario where

$$\text{supp}(\mathbf{p}^t) \cap \left( \cup_{t' \neq t} \text{supp}(\mathbf{p}^{t'}) \right) = \emptyset. \qquad (60)$$

One example where (60) holds is when $\text{supp}(\mathbf{p}^t) \subseteq [s/2]$ and $\text{supp}(\mathbf{p}^{t'}) \subseteq \{s/2 + 1, \ldots, d\}$ for all $t' \neq t$. If (60) holds, then the support of the datapoints generated by $\{\mathbf{p}^{t'} : t' \neq t\}$ does not

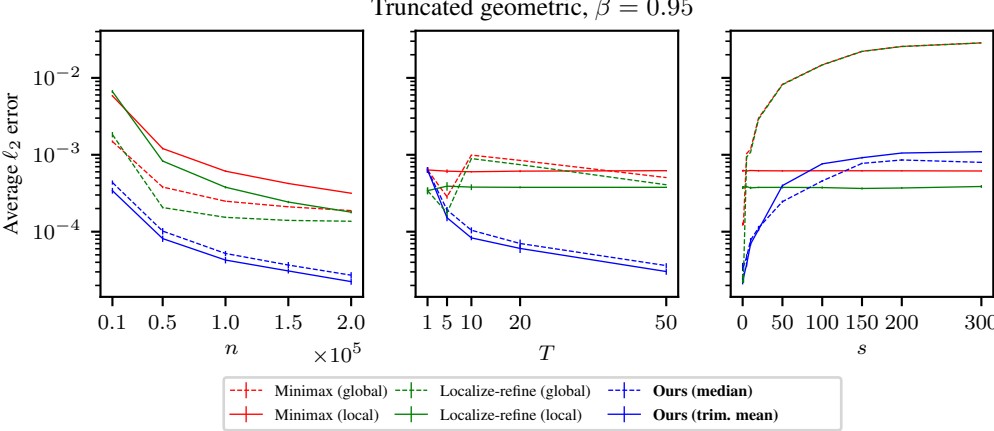

Figure 3: Average $\ell_2$ estimation error in synthetic experiment using the truncated geometric distribution. (Left): Fixing $s = 5$, $T = 30$ and varying $n$. (Middle): Fixing $s = 5$, $n = 100,000$ and varying $T$. (Right): Fixing $T = 30$, $n = 100,000$ and varying $s$. The standard error bars are obtained from 10 independent runs.

overlap with the support of those generated by $\mathbf{p}^t$, and hence former are not informative for estimating $\mathbf{p}^t$. Therefore, by further combining (59) with the existing lower bound result [14, Thm 2] for the $s/2$-sparse homogeneous setup, where all datapoints are generated by a single $s/2$-sparse distribution, that for any estimation method (possibly based on interactive encoding),

$$
\inf_{\substack{(W^{t,[n]})}} \sup_{\substack{\|\mathbf{p}^t\|_0 \leq s/2 \\ \mathbf{p}^t \in \mathcal{P}_d}} \mathbb{E}[\|\widehat{\mathbf{p}}^t - \mathbf{p}^t\|_q^q] = \Omega\left( (s/2)^{1-q/2} \left( \frac{\max\{2^b, s/2\}}{2^b n} \right)^{q/2} \right)
$$

$$
= \Omega\left( s^{1-q/2} \left( \frac{\max\{2^b, s\}}{2^b n} \right)^{q/2} \right).
$$

Thus, we have

$$
\inf_{\substack{(W^{t',[n]})_{t' \in [T]} \\ \widehat{\mathbf{p}}^t}} \sup_{\substack{\mathbf{p}^\star \in \mathcal{P}_d \\ \{\mathbf{p}^{t'} : t' \in [T]\} \subseteq \mathbb{B}_s(\mathbf{p}^\star)}} \mathbb{E}[\|\widehat{\mathbf{p}}^t - \mathbf{p}^t\|_q^q] \geq \inf_{\substack{(W^{t',[n]})_{t' \in [T]} \\ \widehat{\mathbf{p}}^t}} \sup_{\substack{\|\mathbf{p}^{t'}\|_0 \leq s/2 \\ \forall\, t' \in [T]}} \mathbb{E}[\|\widehat{\mathbf{p}}^t - \mathbf{p}^t\|_q^q]
$$

$$
\geq \inf_{\substack{(W^{t',[n]})_{t' \in [T]} \\ \widehat{\mathbf{p}}^t}} \sup_{\substack{\|\mathbf{p}^{t'}\|_0 \leq s/2,\, \forall\, t' \in [T] \\ (60)\text{ holds}}} \mathbb{E}[\|\widehat{\mathbf{p}}^t - \mathbf{p}^t\|_q^q] = \inf_{\substack{(W^{t,[n]})}} \sup_{\substack{\|\mathbf{p}^t\|_0 \leq s/2 \\ \mathbf{p}^t \in \mathcal{P}_d}} \mathbb{E}[\|\widehat{\mathbf{p}}^t - \mathbf{p}^t\|_q^q]
$$

$$
= \Omega\left( s^{1-q/2} \left( \frac{\max\{2^b, s\}}{2^b n} \right)^{q/2} \right).
$$

This proves that the lower bound is at least of the order of the first term in (57). Overall, we conclude the desired result.

## F  Supplementary Experiments

**Truncated geometric distribution**   We consider the truncated geometric distribution with parameter $\beta \in (0, 1)$, $\mathbf{p}^\star = \frac{1-\beta}{1-\beta^d}(1, \beta, \ldots, \beta^{d-1})$, as the central distribution and repeat the experiment in Section 5.1. We use $d = 300, \beta = 0.95, b = 2$ and vary $n, T, s$. Figure 3 summarizes the results. As in Section 5.1, we observe that our methods outperform the baseline methods in most cases, especially when $s$ is small. Also, we see the benefit of collaboration, *i.e.*, decreasing trend of the error as $T$ increases, only in our methods.

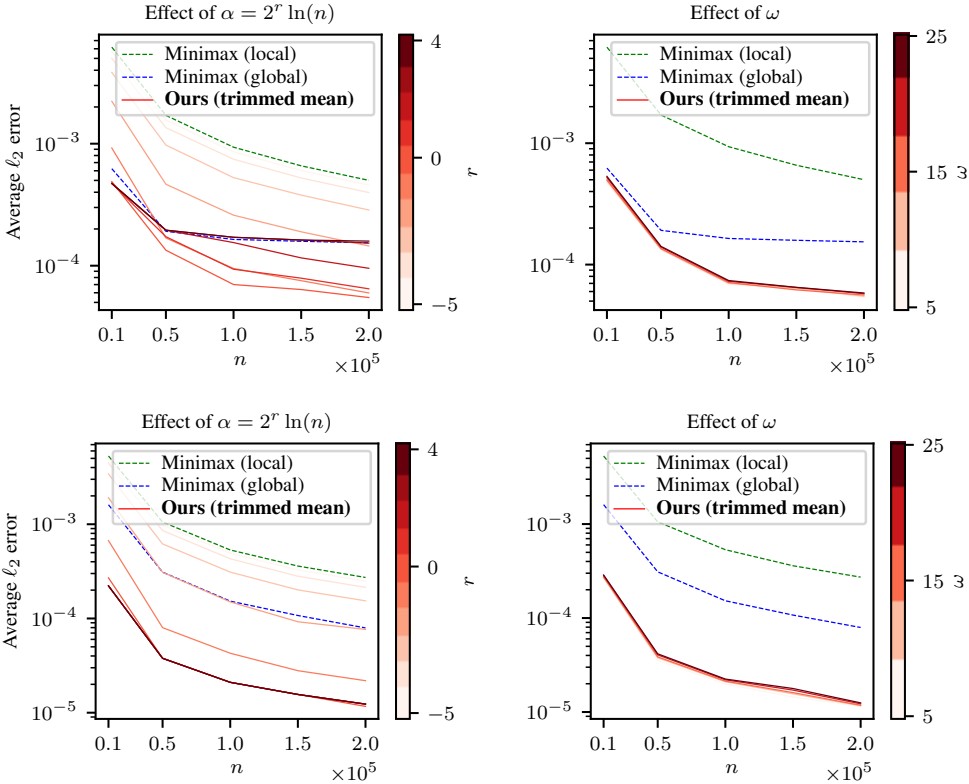

Figure 4: Effect of the hyperparameters $\alpha$ and $\omega$. The top row shows results for the uniform distribution and the bottom row shows the results for the truncated geometric distribution with $\beta = 0.8$.

**Hyperparmeter selection.** We provide additional experiments using different hyperparameters $\alpha$ and $\omega$ from discussed in Section 5.1. All other settings are identical to Section 5.1. We test the hyperparameters $(\alpha, \omega) = (2^r \ln(n), 0.1)$ for $r \in \{-5, -4, \ldots, 4\}$ and $(\alpha, \omega) = (\ln(n), \omega)$ for $\omega \in \{0.05, 0.1, \ldots, 0.25\}$. Figure 4 summarizes the results.

We find that setting the threshold $\alpha$ too small leads to replacing almost all coordinates of the central estimate $\widehat{\mathbf{p}}^\star$ with local ones. In the extreme case of $\alpha \approx 0$, our method is essentially returns the local minimax estimates. On the other hand, we observe that the performance of our method is less sensitive to the trimming proportion $\omega$.

While the choice of $\alpha$ is crucial to the performance of our method, we argue that it is possible to select a reasonably good $\alpha$ by checking the number of fine-tuned entries, *i.e.*,

$$\frac{1}{T} \sum_{t=1}^{T} \left| \left\{ k \in [d] : |[\widehat{\mathbf{b}}^\star]_k - [\widehat{\mathbf{b}}^t]_k| > \sqrt{\frac{\alpha[\widehat{\mathbf{b}}^t]_k}{n}} \right\} \right|.$$

In Figure 5, we observe that more than half ($d/2 = 150$) of the entries are fine-tuned when $r \in \{-5, -4, -3\}$. These correspond to the three curves in the top left of Figure 4 that perform no better than the baseline methods. In conclusion, by selecting $\alpha$ such that the number of fine-tuned entries are small enough compared to $d$, it is possible to reproduce the results in Section 5.

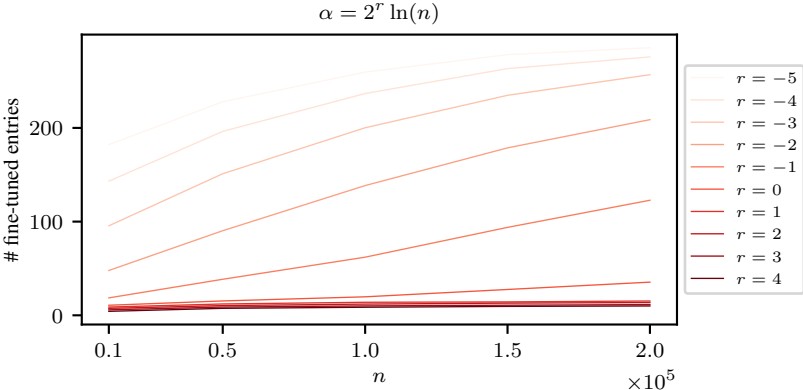

Figure 5: Average number of fine-tuned entries for different values of $\alpha = 2^r \ln(n)$. We use the trimmed mean with $\omega = 0.1$ and the uniform distribution with $d = 300$. This corresponds to the top left of Figure 4.