# OpenReview forum: "Collaborative Learning of Discrete Distributions under Heterogeneity and Communication Constraints"
_NeurIPS.cc/2022/Conference — NeurIPS 2022 Accept_

### Official Review · Reviewer_d6ap · 2022-07-11

**Rating:** 4
**Confidence:** 2
**Soundness:** 3 good
**Presentation:** 3 good
**Contribution:** 2 fair

**Summary:**

This paper proposed a new method for collaborative learning of categorical distribution when the data between users are heterogenous but only sparsely different in a few entries. The methods are composed of two steps: 1. collaborative learning of centralized hashed distribution by robust estimation over multiple agents; 2. fine tuning for each user, which is executed through threshold plus projection. The theoretical part of the paper shows the methods work well when there are lots of users and the difference are sparse. Futher, it states the median-based method is minimax optimal

**Questions:**

1. In theorem 1, how to interpret the notation $n=\Omega(2^b \ln(n))$ that has variable $n$ at both sides? $\tilde{O}(\cdot)$ there hide logarithmic factors with respect to which variables (should be datapoints $n$)? how to choose $\alpha$ in practice?

2. The proposed algorithm seems to be one-shot, i.e. processing all data at once. While in the reviewer's experience, the learning scenario is more common to be iterative or online style. That is the users or agents may continuously send one or multiple data to the central server. Is it worth to modify the algorithm to be caple of accepting the stream data or just iteratively updating the style? Or can authors provide more application cases when this batch-style processing is preferred?

3. In the proof of proposition 2, it states that ".., it holds for $q=1, 2, \mathbb{E}\\|x-x^\star\\|^q_q$ = ...". The first line of proof wrote
"$\mathbb{E}\\|x-x^\star\\|^q_q = \sum_{k\in[d]} \mathbb{E} [|{\rm Proj}(...) - {\rm Proj}(...)|^q]$", i.e., expanding to each entry. Why does it hold for $q=2$ case?


**Strengths And Weaknesses:**

Originality: The idea of the methods and the problem setting don't sound very novel to me. But the proof and theoretical part look

Quality: The proof is mainly built upon the classical statistical framework, which looks solid to me.

Clarity:  From motivation to setup, from algorithm description to theoretical analysis, from notation to experiment setting, the paper is written in a good shape so that I think the readers can quickly grasp the core idea and technique.

Significance: I felt the scope of usage and problem setting are more or less limited. But I am conservative on this opinion since I may miss some important applications in practice.

---

> ### Author Response · Authors · 2022-08-02
> **Response to Reviewer d6ap (Part II)**
>
> > "In theorem 1, how to interpret the notation $n=\Omega(2^b\ln(n))$ that has variable $n$ at both sides? $\tilde{O}(\cdot)$ there hide logarithmic factors with respect to which variables (should be data points $n$)? how to choose $\alpha$ in practice?"
>
> For $n=\Omega(2^b\ln(n))$, the precise condition we need in the analysis is $n\geq 2^{b+6}\ln(n)$. In $\tilde{O}(\cdot)$, we hide logarithmic factors with respect to $n$ and $T$. We apologize for omitting the details on the asymptotic notations in the paper. We have supplemented these details and clarified the confusions in Theorem 1 and 2 (see Section 3.2) in the revision.
>
> In practice, we have provided a heuristic strategy in selecting a reasonably good $\alpha$ by checking the number of fine-tuned entries (see Appendix F).
>
> > "The proposed algorithm seems to be one-shot, i.e. processing all data at once. While in the reviewer's experience, the learning scenario is more common to be iterative or online style. That is the users or agents may continuously send one or multiple data to the central server. Is it worth to modify the algorithm to be caple of accepting the stream data or just iteratively updating the style? Or can authors provide more application cases when this batch-style processing is preferred?"
>
> We understand the need of leveraging stream data as suggested by reviewer. In fact, our SHIFT method can be easily modified to receive stream data.
> In the SHIFT method, the key logic is to fine-tune the local estimators, each of which is obtained by averaging the hashed samples of cluster (see Appendix A). The fine-tuning procedure and the entry-wise robust estimates only take a few sweeps of the local estimators, which is computationally cheap. When a new datapoint comes in, the local estimators can be updated incrementally due to its averaging structure. Specifically, when the hashed version of the $(n+1)$-th datapoint $X^{t,n+1}$ of cluster $t$ is received by the server,  the according local estimator can be updated as follows:
> $$\check{\mathbf{b}}^t\\;\leftarrow\\;\frac{n}{n+1}\check{\mathbf{b}}^t+\frac{1}{n+1}\mathrm{decode}(\mathrm{hash}_\text{encode}(X^{t,n+1}))$$
> where $\mathrm{decode}$ and $\mathrm{hash}_\text{encode}$ indicate the hash-based encoding procedure and the decoding procedure respectively (see Appendix A for more details)
>
> In this way, the server can keep the local estimators which can be updated in an online fashion, and output the fine-tuned estimators whenever needed.
>
> > "In the proof of proposition 2, it states that ".., it holds for
> $q=1,2\, \mathbb{E}\Vert x-x^\star\Vert\_q^q=...$".
> The first line of proof wrote "$\mathbb{E}\Vert x-x^\star\Vert_q^q=\sum\_{k\in[d]}\mathbb{E}[| \mathrm{Proj}(\dots)-\mathrm{Proj}(\dots)|^q]$", i.e., expanding to each entry. Why does it hold for $q=2$ case?"
>
> We apologize for making two typos in the last two terms in line 518. The typos should be fixed as
> $\mathbb{E}[\Vert \mathbf{y}-\mathbf{y}^\star\Vert\_q]\\; \rightarrow\\;\mathbb{E}[\Vert \mathbf{y}-\mathbf{y}^\star\Vert\_q^q]$ and $O(\mathbb{E}[\Vert\mathbf{y}-\mathbf{y}\Vert\_q^q])\\;\rightarrow\\; O(\mathbb{E}[\|\mathbf{y}-\mathbf{y}^\star\|_q^q])$. To answer the reviewer's question, we re-state our proof for Proposition 2 below for the case where $q=2$.
>
> When $q=2$, the subscript indicates we are using the $\ell_2$ norm. Hence, we have $$\mathbb{E}[\Vert\mathbf{x}-\mathbf{x}^\star\Vert\_q^q]=\mathbb{E}[\Vert\mathbf{x}-\mathbf{x}^\star\Vert_2^2]=\mathbb{E}\left[\sum_{k\in[d]}\left(x\_k-x\_k^\star\right)^2\right]=\sum\_{k\in[d]}\mathbb{E}\left[\left(x\_k-x\_k^\star\right)^2\right]$$
> where $x\_k$ is the $k$-th entry of $\mathbf{x}$ and the same notation applies to $\mathbf{x}^\star$ too. By following the definition of $\mathbf{x}$ and $\mathbf{x}^\star$, we can reach
> \begin{align}
> \sum\_{k\in[d]}\mathbb{E}\left[\left(x\_k-x\_k^\star\right)^2\right]=&\sum\_{k\in[d]}\mathbb{E}\left[\left|\mathrm{proj}\_{[0,1]}\left(\frac{2^b y\_k-1}{2^b-1}\right)-\mathrm{proj}_{[0,1]}\left(\frac{2^b y^\star\_k-1}{2^b-1}\right)\right|^2\right]\\\\
> \leq &\sum\_{k\in[d]}\mathbb{E}\left[\left|\frac{2^b( y\_k-y^\star\_k)}{2^b-1}\right|^2\right]= \left(\frac{2^b}{2^b-1}\right)^2\mathbb{E}[\Vert\mathbf{y}-\mathbf{y}^\star\Vert\_q^q]=O(\mathbb{E}[\Vert\mathbf{y}-\mathbf{y}^\star\Vert\_q^q]).
> \end{align}
>
> Regardless of the two typos, the statement of Proposition 2 is correct, and that is what we used in later analysis, i.e., the two typos do **not influence the soundness** of our analysis.
>
> **We will be glad to clarify any further comments. Thanks again!**

---

> > ### Author Response · Authors · 2022-08-02
> > **Response to Reviewer d6ap (Part II, Continued)**
> >
> > #Reference
> >
> > We use the same citation indexes for the papers cited in the manuscript and indicate the additional citations by letter "R" below.
> >
> > [R1] Robert D. Nowak. (2003). Distributed EM algorithms for density estimation in sensor networks. IEEE International Conference on Acoustics, Speech, and Signal Processing.
> >
> > [R2] Minqi Zhou, Heng Tao Shen and Xiaofang Zhou and Weining Qian and Aoying Zhou. (2012). Effective Data Density Estimation in Ring-Based P2P Networks. In International Conference on Data Engineering.
> >
> > [R3] Slavov2013AGA. (2013). A gossip-based approach for Internet-scale cardinality estimation of XPath queries over distributed semistructured data. The VLDB Journal.
> >
> > [R4] Keith Bonawitz, Vladimir Ivanov, Ben Kreuter, Antonio Marcedone, H. B. McMahan, Sarvar Patel, Daniel Ramage, Aaron Segal, Karn Seth. (2016). Practical Secure Aggregation for Federated Learning on User-Held Data. ArXiv.
> >
> > [R5] Dan Wang, Siping Shi, Yifei Zhu, Zhu Han. (2021). Federated Analytics: Opportunities and Challenges. IEEE Network.
> >
> > [R6] Leighton Pate Barnes, Huseyin A. Inan, Berivan Isik,  Ayfer Ozgur. (2020). rTop-k: A Statistical Estimation Approach to Distributed SGD. IEEE Journal on Selected Areas in Information Theory.

---

> ### Author Response · Authors · 2022-08-02
> **Response to Reviewer d6ap (Part I)**
>
> We thank the reviewer for the thoughtful review of our work. We address the main issues raised by the reviewer below:
>
> > "Originality: The idea of the methods and the problem setting don't sound very novel to me. But the proof and theoretical part look"
>
> We thank the reviewer for recognizing our novelty in the analysis and theoretical results. We summarize the novelty of our problem setup, method, and theory as below.
>
> **Novelty of our problem setup and heterogeneity model**
>
> As stated in our introduction, heterogeneity is a common challenge in distributed/federated learning tasks which causes significance performance drops if not
> handled properly. While previous works [7, 15, 16, 27, 28] have studied  communication-constrained estimation of a single distribution from i.i.d. data points, none of them have considered the setup with multiple distributions to be estimated and the according heterogeneity of distributions. Our work is the **first** to study such a multi-task problem with heterogeneity in communication-constrained estimation.
>
> **Novelty of our method**
>
> The proposed SHIFT method exhibits novel ideas: collaborative learning and fine-tuning. The collaborative learning step is implemented by entry-wise robust estimates which, to our knowledge, have not appeared in the communication-constrained estimation literature before. The fine-tuning step is simple, yet novel, as it requires entry-wise thresholds and tuning which have not appeared in previous works on communication-constrained estimation too. The novel entry-wise thresholds are carefully designed to uncover the entry misalignment (sparse heterogeneity) between local distributions and the global one. By doing so, we have proved that the proposed SHIFT method enjoys improved sample efficiency compared with other methods that do not learn distributions from multiple clusters (see the other two methods with "data usage" marked as "separate" in Table 1).
>
> **Novelty of our theory**
>
> The derivation of the lower bound, and proving the rate that matches this lower bound, are highly non-trivial in this new problem.
> In order to guarantee the optimal upper bound of our method, we need to carefully control the error due to decoding, the effect of sparse heterogeneity on robust estimates, the cost of identifying mis-aligned entries, etc. Each aspect requires a sharp and novel analysis. For example, the results of collaborate learning (Lemma 1, Corollary 1) need to be analyzed in an entry-wise manner.
> This is different from the related analysis techniques in the literature [32, 38, 39] on robust estimates dealing with outliers. The overall proof of our method requires a new combination of ideas from robust estimation, sparse learning, and communication-constrained information processing, which, to our knowledge, has not appeared in the literature.
>
>
>
>
>
>
>
>
>
>
>
>
>
> > "Significance: I felt the scope of usage and problem setting are more or less limited. But I am conservative on this opinion since I may miss some important applications in practice."
>
>
> We will explain the motivation from two perspectives: (1) importance in federated learning and federated analytics, (2) several real world applications that require distributed statistical estimation as key building block
>
>
> **1. Importance in federated learning and federated analytics**
>
> - While many works in the FL literature mainly focus on learning predictive models, it is part of our motivation that learning the distribution itself could be another important task.
> A number of works in ML venues [3, 4, 14, 15, 16] have already studied similar problems.
> We also refer the reviewer to Section 3.5 and Section 4.4.3 of this survey paper [31] and references therein for further review of papers in distributed statistical estimation.
>
> - Federated analytics [R4, R5] aims to perform statistical analysis on raw data that are stored locally on users’ devices.
> For example, it includes computing basic statistics such as mean, quantile, frequency, etc.
> Generally speaking, it can be understood as estimating distribution of locally stored data.
> While the privacy is a main concern in federated analytics research, communication is also a central issue in large-scale applications. In this regard, federated analytics can be a potential application of our work.

---

> > ### Author Response · Authors · 2022-08-02
> > **Response to Reviewer d6ap (Part I, Continued)**
> >
> > **2. Real world applications**
> >
> > - Our problem setup can be naturally generalized to continuous density estimation tasks. In various real data applications, it is required to reconstruct the data distribution from scattered measurements. Examples include sensor networks and P2P (Peer2Peer) systems, load balancing, query processing (see, e.g., [R1, R2, R3] and references therein). Specifically in P2P systems, by knowing the data distribution, data can be dispersed much more evenly among the peers. Different query types in P2P systems,
> > such as range queries, aggregation queries and skyline
> > queries, can also be more efficiently supported with the
> > information on data distribution.
> >
> > - Distributed learning algorithms require gradient aggregation from all local users/datasets/machines where the associated communication can be an overhead. The aggregation of ***stochastic*** local gradients can be naturally viewed as a statistical estimation model. As demonstrated in [R6], the analyses and results of communication-constrained estimation can be easily adapted to obtain new insights in the design of communication-efficient training techniques.

---

> ### Author Response · Authors · 2022-08-07
> **Could you please confirm whether our response has addressed your questions?**
>
> Dear reviewer d6ap,
>
> Thanks very much for your careful review. Could you please confirm whether our response have addressed your questions? We are happy to address any further questions or comments.
>
> Best,
> The authors of paper12094

---

### Official Review · Reviewer_yqYc · 2022-07-11

**Rating:** 7
**Confidence:** 3
**Soundness:** 4 excellent
**Presentation:** 4 excellent
**Contribution:** 3 good

**Summary:**

The paper proposes a federated algorithm for efficiently estimating local data distributions with hashed messages to reduce communication costs, provided that there exists a central/global distribution such that the local ones differ from the global one in a number of bins/entries much smaller than the support size. The approach relies on a robust estimate of the global mass of an entry and it is shown to be minimax optimal wrt the estimation error of the local distributions when using the median estimator. The key intuition behind the approach is that the entries that vary the less between users are probably those that are homogeneous.
An empirical analysis both on synthetic and real datasets illustrate in which settings the method improves on global and local baselines, in particular when the communication budget b is sufficiently large.

**Questions:**

1. Why does the encoding function $W^{t, j}$ depend also on the data point? In which cases is this function not given per user?
2. How difficult would it be to extend the current results to continuous distributions?

**Limitations:**

The principal limitation of the work is in not considering privacy of the studied system. Even if it is out of the scope of the paper, it would be helpful to mention possible tools when working with sensitive data.

**Strengths And Weaknesses:**

The paper is well written. I cannot judge the novelty of the results as I am not familiar with the literature on distributed learning of discrete distributions.

I have a few minor comments on the presentation:
1. In the introduction, a good motivation is given for federated learning but not for learning data distributions in this context. It would be interesting to report some practical examples where it is more useful to estimate the local data distributions instead of local predictors, as the former has probably a higher privacy cost than the latter. Providing a stronger motivation would allow to better asses the impact of the contributions.
2. It should be made clear from the beginning that only the problem of learning discrete distributions is tackled in the current work.
3. When citing the works on federated multi-task learning, the following seminal work should be cited as well:
- Smith, Virginia, et al. "Federated multi-task learning." Advances in neural information processing systems 30 (2017).

Specific remarks and typos:
1. line 28: Is performance drop observed in terms of time convergence or accuracy?
2. line 79: What is it meant by shared randomness here?
3. line 189: $n = \Omega()$ -> $\alpha = \Omega()$?
4. line 243: rest -> remaining
5. line 269: What is it meant as prior in this context?
6. line 304: sparsity -> sparse heterogeneity
7. Figure 2: Why does the error of the proposed approach with the median estimator improve between s = 200 and s = 300?

---

> ### Author Response · Authors · 2022-08-02
> **Response to Reviewer yqYc (Part III)**
>
> > "How difficult would it be to extend the current results to continuous distributions?"
>
> In [27] it is shown that one can optimally estimate a non-parametric but smooth density by binning it into a discrete distribution with an bandwidth $h^* = (n 2^b)^{-\frac{1}{2(s + 1)}} \vee n^{-\frac{1}{2s + 1}}$. Here $b$ is the communication bits, $n$ is the sample size, and $s$ is the Holder smoothness of the unknown density.
> Although we have a heterogeneous multi-task setting, we believe a similar extension would be possible by assuming smoothness on the densities and choosing binning schemes properly.
>
> > "The principal limitation of the work is in not considering privacy of the studied system. Even if it is out of the scope of the paper, it would be helpful to mention possible tools when working with sensitive data."
>
> We thank the reviewer for emphasisizing the importance of privacy. While not studied in this work, we believe our high-level idea of collaborative learning + fine-tuning can be combined with existing privacy-constrained estimation techniques in [3, 4], e.g., adding appropriately noises, to boost the performance. We think our work can be the first step to establish the optimality in the privacy-constrained, heterogeneous scenario (as in [3, 4]) in future work.
>
> **We thank the reviewer for the helpful comments. We are more than happy to have more discussion.**
>
> #Reference
>
> We use the same citation indexes for the papers cited in the manuscript and indicate the additional citations by letter "R" below.
>
> [R1] Robert D. Nowak. (2003). Distributed EM algorithms for density estimation in sensor networks. IEEE International Conference on Acoustics, Speech, and Signal Processing.
>
> [R2] Minqi Zhou, Heng Tao Shen and Xiaofang Zhou and Weining Qian and Aoying Zhou. (2012). Effective Data Density Estimation in Ring-Based P2P Networks. In International Conference on Data Engineering.
>
> [R3] Slavov2013AGA. (2013). A gossip-based approach for Internet-scale cardinality estimation of XPath queries over distributed semistructured data. The VLDB Journal.
>
> [R4] Keith Bonawitz, Vladimir Ivanov, Ben Kreuter, Antonio Marcedone, H. B. McMahan, Sarvar Patel, Daniel Ramage, Aaron Segal, Karn Seth. (2016). Practical Secure Aggregation for Federated Learning on User-Held Data. ArXiv.
>
> [R5] Dan Wang, Siping Shi, Yifei Zhu, Zhu Han. (2021). Federated Analytics: Opportunities and Challenges. IEEE Network.
>
> [R6] Leighton Pate Barnes, Huseyin A. Inan, Berivan Isik,  Ayfer Ozgur. (2020). rTop-k: A Statistical Estimation Approach to Distributed SGD. IEEE Journal on Selected Areas in Information Theory.
>
> [R7] Sai Praneeth Karimireddy, Satyen Kale, Mehryar Mohri, Sashank J. Reddi,Sebastian U. Stich,Ananda Theertha Suresh. (2020). SCAFFOLD: Stochastic Controlled Averaging for Federated Learning. In International Conference of Machine Learning.

---

> > ### Comment · Reviewer_yqYc · 2022-08-08
> > **confirm initial score**
> >
> > I thank the authors for their detailed response in particular on the motivation. I think this discussion should be reported in the main text.
> > After reading also the other reviews, I decided to keep my initial recommendation of accepting the paper.

---

> > > ### Author Response · Authors · 2022-08-08
> > > **Many thanks for maintaining the recommendation of acceptance. We really appreciate it!**
> > >
> > > \
> > > Thank you very much for your decision and the valuable comments. Since the current page limit does not allow adding much content, we will definitely include the discussion in the later revision (one more page will be allowed once the paper gets accepted).

---

> ### Author Response · Authors · 2022-08-02
> **Response to Reviewer yqYc (Part II)**
>
> > "line 79: What is it meant by shared randomness here?"
>
> While the hash mappings are generated locally, the server can access the randomness of the hashing mappings in the decoding stage. In other words, the randomness of each hash mapping is shared between the local user where it was generated and the server.
> Specifically, in our paper, the server needs to use the randomness to calculate $N\_k^t= |\\{j ∈ [n]:\\,h^{t,j} (k) = Y^{t,j}\\}|$. Here $h^{t,j}$ is the hash mapping associated with data point $X^{t,j}$, and $Y^{t,j}$ is the hashed value of $X^{t,j}$.
>
> We also remark that the shared randomness exists in most hashing-based communication mechanisms and has appeared extensively in communication-constrained estimation literature [2, 3, 4, 15, 16, 23, 27].
>
> > line 189: $n=\Omega(\cdot)\;\rightarrow \alpha = \Omega()$?
>
> $n=\Omega(2^b\ln(n))$ here should be replaced with the more precise $n\geq 2^{b+6}\ln(n)$. That is what we need for the guarantee. For the condition on $\alpha$, we require $\alpha$ to satisfy $c\ln(n)\leq \alpha\leq C\ln(n)$ where $c$ is a real number revealed by our analysis (see Theorem 5 in Appendix C) and $C$ $(C\geq c)$ can be any number not dependent on $n$, $s$, $d$, $b$. The insight for choosing $\alpha$ is that a small $\alpha$ selects too many entries to be estimated locally and a large $\alpha$ cannot effectively identify the positions of misaligned entries where $[\mathbf{p}^t]_k\neq [\mathbf{p}^\star]_k$.
>
> We thank the reviewer for the comments on notation. We have clarified the related notations in Theorem 1 and 2 (see Section 3.2) in the revision (in blue).
>
> > "line 269: What is it meant as prior in this context?"
>
> This is a technical term in the field of statistics. When proving a minimax lower bound, we care about quantities taking a form like $\inf\_{\hat{\mathbf{p}}}\max\_{\mathbf{p}\in\mathcal{P}\_d}\mathrm{Measure}(\hat{\mathbf{p}},\mathbf{p})$ where $\mathbf{p}$ is an unknown parameter lying in domain $\mathcal{P}\_d$ and $\hat{\mathbf{p}}$ can be arbitrary estimator (algorithm). Sometimes explicitly identifying the worst value $\mathbf{p}^{worst}$ that maximizes the inner maximization can be hard, *i.e.,* $\inf\_{\hat{\mathbf{p}}}\mathrm{Measure}(\hat{\mathbf{p}},\mathbf{p}^{worst})=\inf\_{\hat{\mathbf{p}}}\max\_{\mathbf{p}\in\mathcal{P}\_d}\mathrm{Measure}(\hat{\mathbf{p}},\mathbf{p})$. Therefore, a standard  technique in statistical decision theory is to find a surrogate distribution $D$ over the domain $\mathcal{P}\_d$ for the unknown parameter $\mathbf{p}$. The distribution for the parameter is also called a **prior** in statistics.
> Since we naturally have
> $$\inf\_{\hat{\mathbf{p}}}\max\_{\mathbf{p}\in\mathcal{P}\_d}\mathrm{Measure}(\hat{\mathbf{p}},\mathbf{p})\geq \inf\_{\hat{\mathbf{p}}}\max\_{\mathrm{prior}\\,D,\\;\mathbf{p}\sim D}\mathrm{Measure}(\hat{\mathbf{p}},\mathbf{p}),$$
> one can lower bound the original minimax quantity by explicitly identifying a worst **prior** for parameter $\mathbf{p}$ instead of a single worst value.
>
> Back to our paper, we have multiple local parameters $\\{\mathbf{p}^t:t\in[T]\\}$ to be estimated. In the $s/2$-sparse case, we will construct a corresponding prior $D^t$ for each parameter $\mathbf{p}^t$. Here "independent priors" means that $D^1,\dots,D^T$ are independent distributions. This intuitively reflects that there is no knowledge towards the relation between the local parameters $\\{\mathbf{p}^t:t\in[T]\\}$ before looking the samples.
>
> We hope the above explanation can help answer the reviewer's question. We are more than happy to clarify any further comments.
>
> > "line 243: rest -> remaining"
>
> > "line 304: sparsity -> sparse heterogeneity"
>
> We have fixed them in our revision.
>
> > "Figure 2: Why does the error of the proposed approach with the median estimator improve between s = 200 and s = 300?"
>
> The parameter $[\mathbf{p}^\star]_k$ is identifiable only when less than a half of local distributions $\mathbf{p}^t$ are misaligned with $\mathbf{p}^\star$ at their $k$-th entry.
> If $s > \frac{d}{2}$, then by the pigeonhole principle, the aforementioned property is violated. So it is beyond the scope of our theoretical predictions, though we included $s > \frac{d}{2}$ in our experiments for completeness.
>
> > "Why does the encoding function $W^{t, j}$ depend also on the data point? In which cases is this function not given per user?"
>
> Some related works in communication-constrained estimations [16, 27] consider a larger family of communication mechanisms that allow sequential interaction. In this case, the encoding function $W^j$ of the $j$-th data point can depend on all previous sent messages
> $Y^{1},\dots,Y^{j-1}$ and thus takes a form like $Y^j=W^j(X^j|Y^{1},\dots,Y^{j-1}).$
> We remark that even though our SHIFT method does not require such sequential interaction, it is still minimax optimal among this larger family of communication mechanisms (see the lower bound definition in Theorem 3, 4).

---

> ### Author Response · Authors · 2022-08-02
> **Response to Reviewer yqYc (Part I)**
>
> We thank the reviewer for the positive comments. We have attempted to address all the questions. We will be glad to clarify any further comments.
>
> > "In the introduction, a good motivation is given for federated learning but not for learning data distributions in this context. It would be interesting to report some practical examples where it is more useful to estimate the local data distributions instead of local predictors, as the former has probably a higher privacy cost than the latter. Providing a stronger motivation would allow to better asses the impact of the contributions."
>
> We will explain the motivation from two perspectives: (1) importance in federated learning and federated analytics, (2) several real world applications that require distributed statistical estimation as a key building block:
>
>
> **1. Importance in federated learning and federated analytics**
>
> - While many works in the FL literature mainly focus on learning predictive models, it is part of our motivation that learning the distribution itself could be another important task.
> A number of works in ML venues [3, 4, 14, 15, 16] have already studied similar distribution estimation problems.
> We also refer the reviewer to Section 3.5 and Section 4.4.3 of this survey paper [31] and references therein for further review of papers in distributed statistical estimation.
>
> - Federated analytics [R4, R5] aims to perform statistical analysis on raw data that are stored locally on users’ devices.
> For example, it includes computing basic statistics such as mean, quantile, frequency, etc.
> Generally speaking, it can be understood as estimating the generating distribution of locally stored data.
> While the privacy is a main concern in federated analytics research, communication-efficiency is also a central issue in large-scale applications. In this regard, federated analytics can be a potential application of our work.
>
> **2. Real world applications**
>
> - As we respond to "the extension to continuous distributions" below, our problem setting can be generalized to continuous density estimation tasks. In various real-world applications, it is required to reconstruct the data distribution from scattered measurements. Examples include sensor networks and P2P (Peer2Peer) systems, load balancing, query processing (see, e.g., [R1, R2, R3] and references therein). Specifically in P2P systems, by knowing the data distribution, data can be dispersed much more evenly among the peers. Different query types in P2P systems,
> such as range queries, aggregation queries, and skyline
> queries, can also be more efficiently supported with the
> information on the data distribution.
>
>
> - Distributed training algorithms require gradient aggregation from all local users/datasets/machines where the associated communication can be an overhead. The aggregation of ***stochastic*** local gradients can be naturally viewed as a statistical estimation model. As demonstrated in [R6], the analyses and results of communication-constrained estimation can be easily adapted to obtain new insights in the design of communication-efficient training techniques.
>
> We hope our explanation can alleviate the reviewer's concern about the motivation of our paper. We will merge the above content to our later revision. For the privacy concern, please see our answer below.
>
> > "It should be made clear from the beginning that only the problem of learning discrete distributions is tackled in the current work."
>
> We mentioned it twice in the abstract and also in the second paragraph of introduction section. We have added the keyword "discrete" to the title in the revision.
>
> > "When citing the works on federated multi-task learning, the following seminal work should be cited as well:
> * Smith, Virginia, et al. "Federated multi-task learning." Advances in neural information processing systems 30 (2017)."
>
> Thanks for pointing out this paper, we have included this paper in the revision.
>
> > "line 28: Is performance drop observed in terms of time convergence or accuracy?"
>
> Generally speaking, the performance drop caused by heterogeneity depends on the metric and task. For example, in Federated Learning, when the heterogeneity of local data is not handled properly, algorithms like FedAvg [37] suffer from the so-called "client-drift phenomenon". This leads to drops both in convergence and test accuracy (see Figure 3 and Table 5 in [R7]).
>
> Particularly, in the problem setup of our paper, the performance drop is in terms of estimation error. Please check Table 1 for the other two methods with data usage marked as "pool".

---

### Official Review · Reviewer_6d3k · 2022-07-11

**Rating:** 7
**Confidence:** 4
**Soundness:** 1 poor
**Presentation:** 2 fair
**Contribution:** 3 good

**Summary:**

The paper considers the estimation of d-dimensional distributions in a decentralized setting. Agents collaborate by sending messages to a central server, but their messages are limited to k bits and $2^k << d$. This setting was already studied by several authors, but the originality here is to allow sparse heterogeneity among agents.

The assumption is that there are $T$ distinct  but unknown d-dimensional distributions, and each agent is associated with one of the $T$ distributions. The set of agents is therefore partitioned into $T$ classes and, as far as I understand, the classes are known to the central server. Each class is characterized by a (small) set of components that have a distribution specific to it. All agents have exactly n samples drawn from their distribution. They send n messages. The ith message is a binary hash of the ith sample of length $k$.

The approach is the following. After receiving all messages, the server first performs a robust estimation of each component of the distributions, considering all the messages sent by all agents. Then it estimates for each cluster the components that are specific to a cluster and replace the global estimate by an estimate computed with the messages received by all agents in the same cluster.

The main results state that this 2 steps procedure is an upper and a lower bound on the estimation error (the l2 norm of the difference of the estimation and the true distribution) for each class. The upper bound contains a term that can be $T$ times smaller because of the collaboration across classes (possible for the estimation of the non-specific components). Another term depends on the sparsity value s rather than the dimension d for the specific components.

While the proof can be a bit technical, the result is not surprising. The proof has essentially to control the weight of the errors due to the decoding of the hash functions in the restricted communication setting.



**Questions:**

1. In the main text the authors use the notation $n = \Omega(2^b ln(n))$ which is clearly false (in the common acceptation of this asymptotic notation $\Omega$, the Landau notation). In the appendix, it is written that $n$ should be larger than $2^{b+6} ln(n)$. It should be written like that in the main text.

   But more importantly, the proof itself in appendix C1 starts considering  two cases that also misuse the Landau notation (Using $O$ and $\Omega$).

2. The setting should be more precise about the central server's knowledge of the problem and the agents. In particular, is the central server able to identify the cluster membership of the clients, the value of $T$, of $s$,...?

3. In the lower bound part, two extreme cases are considered (homogeneous and s/2-sparse) and I don't understand why the lower bound can be expressed as the sum of these two cases (and not for instance the max of the two...)


**Limitations:**

As explained above, the setting should be more precise about the central server's knowledge because it induces some limitations.

**Strengths And Weaknesses:**


# Strengths
- The setting is interesting and extends known results
- There are original contributions in the proof
- some experiments give an empirical evaluation of the proposed algorithm

# Weaknesses
1. The statement of the main theorems are not clear enough in the main text
2. I also recommend clarifications in the exposition of the setting
3. I have not understood the approach used in the lower bound

---

> ### Author Response · Authors · 2022-08-02
> **Response to Reviewer 6d3k (Part II)**
>
> > "The statement of the main theorems are not clear enough in the main text"
>
> > "In the main text the authors use the notation $n = \Omega(2^b \ln(n))$ which is clearly false (in the common acceptation of this asymptotic notation $\Omega$, the Landau notation). In the appendix, it is written that $n$ should be larger than $2^{b + 6} \ln (n)$. It should be written like that in the main text."
>
> We thank the reviewer for pointing out this issue with the use of asymptotic notations in the main text.
> Formal statements of Theorem 1 and 2, without asymptotic notations, can be found in the Appendix (Theorem 5, 6).
> Also, as recommended by the reviewer, we have made our statements of Theorem 1 and 2 more explicit in the revision stated in blue.
>
> However, we would like to emphasize that this slip in notations **does not affect the overall soundness of our main results.**
>
>
> > "But more importantly, the proof itself in appendix C1 starts considering two cases that also misuse the Landau notation (Using $O$ and $\Omega$)."
>
> The main idea behind the proof is to consider two cases: (Case 1) when $T \leq 20\ln(n)$; (Case 2) when $T \geq 20\ln(n)$.
> We emphasize that the proof is already complete and sound since we specified the exact constant in line 633 of the original manuscript (line 637 in the updated manuscript) as $T \geq 20 \ln(n)$.
> We used the $O()$ and $\Omega()$ notations to try to better illustrate the main ideas.
>
> However, we agree with the reviewer that this can be an abuse of notations. We have fixed it in the revision. We hope this minor slip of notations (which we easily fixed) will not become a deal breaker.
>
> > "I have not understood the approach used in the lower bound"
>
> > "In the lower bound part, two extreme cases are considered (homogeneous and s/2-sparse) and I don't understand why the lower bound can be expressed as the sum of these two cases (and not for instance the max of the two...)"
>
> As mentioned by the reviewer, we consider two extreme cases which help to establish the lower bound for the minimax error as follows:
> $$\inf \sup \mathbb{E}[\Vert \hat{\mathbf{p}}^t - \mathbf{p} \Vert_q^q] = \Omega\left( \frac{d}{(2^b T n)^{q / 2}} \right) \quad \text{ (homogeneous case)},$$
> $$\inf \sup \mathbb{E}[\Vert \hat{\mathbf{p}}^t - \mathbf{p} \Vert_q^q] = \Omega\left( s^{1 - q/2} \left( \frac{\max\{2^b, s \}}{2^b n} \right)^{q / 2} \right) \quad  \text{ ($s/2$-sparse case)}$$
> where $q=1$ or $2$.
> Now, the $\Omega()$ notation is used in the standard way: $A = \Omega(B)$ with $A,\\,B>0$ means that there is a universal constant $c>0$ such that $A \ge c B$, for all instances of $A,B$ under consideration.
> Since the supremum is over all possible parameter instances, the minimax error is lower bounded by the estimation error (lower bound) for any specific instance, e.g., the homogeneous instance and the $s/2$-sparse instance.
>
> Then, we use the simple inequality $\max \\{a, b \\} \geq \frac{a + b}{2}$, for positive $a, b$, to derive
> $$\inf \sup \mathbb{E}[\Vert \hat{\mathbf{p}}^t - \mathbf{p} \Vert_q^q] = \Omega\left( \frac{d}{(2^b T n)^{q / 2}} \right) + \Omega\left( s^{1 - q/2} \left( \frac{\max\{2^b, s \}}{2^b n} \right)^{q / 2} \right).$$
>
> Note that, one of the properties of the $\Omega()$ notation is that $\Omega(A+B) = \Omega(\max(A,B))$, i.e., since  $a + b \geq \max \\{a, b \\} \geq \frac{a + b}{2}$ for any $a,\\, b\geq 0$, the maximum and the sum are of the same order. Thus, a lower bound of the order of the maximum and of the order of the sum are equivalent.
>
>
> To our knowledge, this type of argument is quite common in proving a minimax lower bound consisting of more than one terms (e.g. see the proof of Theorem 2 in [R1], or proofs of Theorem 1, 2 in [R2]).
>
> **We again thank the reviewer: If any of your concerns is not addressed here, please let us know. We are more than happy to discuss it further.**
>
>
> #Reference
>
> We use the same citation indexes for the papers cited in the manuscript and indicate the additional citations by letter "R" below.
>
> [R1] Jiaqi Yang, Wei Hu, Jason D. Lee, Simon S. Du. (2020, September). Impact of Representation Learning in Linear Bandits. In International Conference on Learning Representations.
>
> [R2] Yucheng Lu and Christopher De Sa. (2021). Optimal Complexity in Decentralized Training. In International Conference on Machine learning.

---

> ### Author Response · Authors · 2022-08-02
> **Response to Reviewer 6d3k (Part I)**
>
> We thank the reviewer for careful reading of the manuscript and helpful comments. We address main concerns as follows:
>
> > "the result is not surprising. The proof has essentially to control the weight of the errors due to the decoding of the hash functions in the restricted communication setting."
>
> **Novelty and importance of our problem setting**
>
> To the best of our knowledge, our result is the first to theoretically justify the benefit of collaboration in communication-constrained and heterogeneous estimation tasks.
> While previous works [7, 15, 16, 27, 28] have studied the task of communication-constrained estimation of a single distribution from i.i.d. data points, none of them have considered heterogeneity. In this respect, our paper has introduced the  sparse-heterogeneity model for the first time and has argued with motivating examples that this model naturally fits many practical applications.
>
>
> As stated in the Introduction section of our paper, (statistical) heterogeneity is a key challenge in distributed/federated learning tasks.
> Indeed, methods that are designed for i.i.d data can experience significant performance drops in the presence of heterogeneity.
> It is indeed nontrivial to benefit from multiple heterogenous datasets in terms of sample efficiency without suffering from a drop (due to heterogeneity).
>
> **Novelty of our method**
>
> The proposed SHIFT method exhibits novel ideas: collaborative learning and fine-tuning. The collaborative learning part is implemented by entry-wise robust estimates which, to our knowledge, have not previously appeared in the communication-constrained estimation literature. The fine-tuning step is simple yet novel, as it requires entry-wise thresholds and tuning, which have not appeared in previous works on communication-constrained estimation either. The novel entry-wise thresholds are carefully designed to uncover the entry-wise misalignment (sparse heterogeneity) between local and global distributions.
> By doing so, we have proved that the proposed SHIFT method enjoys improved sample efficiency compared with other methods that do not learn distributions from multiple clusters (see the other two methods with "data usage" marked as "separate" in Table 1).
>
>
> **Novelty of our theory and proof techniques**
>
> We would like to emphasize that the derivation of the lower bound, and proving the rate that matches this lower bound, are indeed non-trivial and novel technical contributions of the paper. In order to guarantee the optimality of the error rate obtained by SHIFT, we need to carefully control not just the error due to decoding, but also the effect of sparse heterogeneity on robust estimates, the cost of identifying mis-aligned entries, etc. Each aspect requires a careful analysis, with many novel aspects. For example, the results on the effects of collaboration (Lemma 1, Corollary 1) need to be analyzed in an entry-wise manner. This is different from the related analysis techniques from the literature [32, 38, 39] on robust estimates dealing with outliers (which are not per-coordinate but global).
> To summarize: The overall proof of our method requires a new combination of ideas from robust estimation, sparse learning, and communication-constrained information processing, which, to our knowledge, is new to the literature.
>
> > "I also recommend clarifications in the exposition of the setting"
>
> > "The setting should be more precise about the central server's knowledge of the problem and the agents. In particular, is the central server able to identify the cluster membership of the clients, the value of $T$, of $s$?"
>
> We thank the reviewer for the comments and have revised the manuscript (in blue) accordingly to emphasize the following points:
>
> - We assume that the cluster memberships of users and the number of clusters $T$ are known to the server. We believe that this is a reasonable assumption in many applications where the users' basic information (which determine their cluster membership) such as location/nationality are already known by the server.
>
>   Another possible interpretation of our setting is that each user is holding multiple data points following his own distribution, and the server observes to which user each hashed message belongs.
>
> - Our Algorithm 1, including the choice of threshold parameter $\alpha$, does not require the knowledge of $s$, and it can adapt to any underlying sparse heterogeneity $s$.

---

> ### Author Response · Authors · 2022-08-07
> **Could you please confirm whether our response has addressed your questions?**
>
> Dear reviewer 6d3k,
>
> Thanks very much for your careful review. Could you please confirm whether our response have addressed your questions? We are happy to address any further questions or comments.
>
> Best,
> The authors of paper12094

---

### Author Response · Authors · 2022-08-06
**Can we have your response to our rebuttals?**

Dear reviewers,

\
Thank you very much for your careful review and valuable feedback. We have addressed all your questions. Could you please respond to our rebuttals? We are happy to clarify any further comments and questions.

\
Best,

The Authors of Paper12094

---

### Meta-Review · Area_Chair_A89N · 2022-08-26

**Recommendation:** Accept
**Confidence:** Certain

**Metareview:**

With 4, 7, and 7, the paper got a wide range of scores. The reviewer who assigned score 4 did not identify any clear flaws in the paper, and did not engage with the authors or other reviewers. Hence I put less weight on the lower review and recommend accepting the paper.

**Award:**

No

---

### Decision · Program_Chairs · 2022-09-14

Accept